# Pretreatment with *Citrus reticulata* ‘Chachi’ Polysaccharide Alleviates Alcohol-Induced Gastric Ulcer by Inhibiting NLRP3/ASC/Caspase-1 and Nrf2/HO-1 Signaling Pathways

**DOI:** 10.3390/nu17132062

**Published:** 2025-06-20

**Authors:** Huosheng Liang, Yiyao Liang, Lipeng Wu, Long Lin, Yunan Yao, Jinji Deng, Jiepei Xu, Huajian Li, Fangfang Gao, Wenlong Xing, Meng Yu, Xuejing Jia, Minyan Wei, Chuwen Li, Guodong Zheng

**Affiliations:** 1School of Biomedical and Pharmaceutical Sciences, Guangdong University of Technology, Guangzhou 510006, China; 2Guangzhou Municipal and Guangdong Provincial Key Laboratory of Molecular Target & Clinical Pharmacology, the NMPA and State Key Laboratory of Respiratory Disease, School of Pharmaceutical Sciences and Affiliated Traditional Chinese Medicine Hospital, Guangzhou Medical University, Guangzhou 511436, China; 3Phase I Clinical Trial Center, Guangzhou Eighth People’s Hospital, Guangzhou Medical University, Guangzhou 510440, China; 4Department of Pharmacy, Guangzhou Panyu Central Hospital, Guangzhou 511400, China; 5Department of Pharmacy, Guangzhou Development District Hospital, Guangzhou 510730, China; 6College of Food Science and Technology, Guangdong Ocean University, Zhanjiang 524088, China

**Keywords:** *Citrus reticulata* ‘Chachi’ polysaccharide, Nrf2/HO-1 pathway, NLRP3 inflammasome, tight junction, alcohol-induced gastric ulcer

## Abstract

Objectives: This study was designed to investigate the gastroprotective effects of *Citrus reticulata* ‘Chachi’ polysaccharide (CRP) against alcohol-induced gastric ulcers (GUs) and to elucidate its underlying mechanisms. Methods: CRP was extracted, purified, and structurally characterized. BALB/c mice (50/250 mg/kg CRP) and GES-1 cells (1 mg/mL CRP) were subjected to alcohol-induced injury. Oxidative stress (SOD, MDA, ROS), inflammation (TNF-α, IL-1β, NLRP3 inflammasome), mucosal barrier proteins (ZO-1, occludin, Claudin-5), and Nrf2/HO-1 signaling were analyzed via histopathology, Western blot, flow cytometry, and immunohistochemistry. Results: CRP pretreatment significantly alleviated gastric lesions, decreased oxidative stress, and suppressed inflammatory responses in alcohol-induced mice. Mechanistically, CRP induced the Nrf2/HO-1 antioxidant pathway while inhibiting the activation of the NLRP3 inflammasome. CRP also restored tight junction protein expression, enhanced mucosal repair, and reduced epithelial apoptosis. In vitro, CRP promoted cell proliferation, migration, and survival of GES-1 cells under alcohol stress. Conclusions: CRP mitigated alcohol-induced GU via dual antioxidant, anti-inflammatory, and barrier-protective mechanisms, positioning it as a considerable agent for GU.

## 1. Introduction

Gastric ulcer (GU), characterized by localized erosive necrosis of the gastric mucosa, represents a significant global health burden, with an estimated prevalence exceeding 10% worldwide [1,2]. Under physiological conditions, the gastric mucosal barrier maintains homeostasis through multilayered defense mechanisms [3]. Exogenous aggressors (e.g., alcohol, NSAIDs, tobacco, and Helicobacter pylori infection) disrupt these defenses, triggering oxidative stress and inflammatory cascades [4,5,6]. Epidemiological studies reveal a rising incidence of alcohol-associated GU, driven by multifaceted molecular mechanisms, posing significant public health challenges [7,8].

Alcohol induces mucosal damage via dual pathways: direct disruption of membrane integrity and NADPH oxidase-mediated ROS overproduction [9]. When ROS accumulation overwhelms endogenous antioxidant systems including SOD and CAT, it induces mitochondrial dysfunction, DNA oxidative damage, and lipid peroxidation (evidenced by MDA production). This oxidative stress cascade, coupled with the activation of key pro-inflammatory signaling pathways, facilitates the robust release of pro-inflammatory cytokines (TNF-α, IL-1β, IL-6), ultimately driving mucosal injury and impairing healing [10]. Current therapies (prostaglandin analogs, histamine receptor antagonists, and proton pump inhibitors) primarily target acid suppression but face limitations, including high recurrence rates, complications, drug side effects, and antibiotic resistance [11,12].

*Citrus reticulata* ‘Chachi’, a cultivar of *Citrus reticulata* Blanco, indigenous to Xinhui District, Guangdong Province, China, is renowned for its dried pericarp, termed Guang Chenpi (Xinhui peel). As a genuine medicinal material in traditional Chinese medicine, Guang Chenpi is distinguished by its high concentration of bioactive components, such as polysaccharides, flavonoids, and volatile oils [13], which collectively contribute to its therapeutic applications in gastrointestinal disorders [14]. Recent studies have elucidated the gastroprotective mechanisms of its bioactive fractions. For instance, the essential oil of Guang Chenpi synergizes with patchouli oil to suppress gastric acid secretion and attenuate apoptosis in alcohol-induced GU models [15]. In addition, aqueous extracts, especially flavonoids, exhibit considerable antioxidant activity in response to alcohol-induced oxidative stress and ulceration [16]. Notably, emerging evidence highlights the critical role of natural polysaccharides in gastric mucosal protection. Polysaccharides from *Panax ginseng*, *Hericium erinaceus*, *Bletilla striata*, and *Dioscorea opposita* have shown the ability to mitigate gastric injury in alcohol-induced GU rats [17,18,19,20]. However, despite the rich polysaccharide content and documented therapeutic value of Guang Chenpi, the specific gastroprotective effects and underlying mechanisms of its polysaccharide fraction (CRP) remain unexplored. Given the limitations of current therapies and the promising potential of natural polysaccharides, investigating CRP represents a compelling strategy for developing novel complementary or alternative GU interventions.

In this study, the gastroprotective effects and underlying mechanisms of CRP against alcohol-induced GU were investigated in alcohol-induced GU models in vivo and in vitro. Specifically, we hypothesized that CRP exerts protection by concurrently targeting oxidative stress (via the Nrf2/HO-1 pathway), inflammation (via suppressing the NLRP3 inflammasome and pro-inflammatory cytokines), and mucosal barrier integrity (via enhancing tight junctions (TJs) proteins). We provided comprehensive evidence demonstrating that CRP exerted significant protective effects in these models. This study establishes a solid theoretical and experimental foundation for the potential development of CRP as a promising natural agent or functional food ingredient for GU prevention and treatment, contributing to the modernization of Guang Chenpi applications.

## 2. Materials and Methods

### 2.1. Reagents and Materials

The pericarp of *Citrus reticulata* ‘Chachi’ used in this study was procured from Xinbaotang Biological Technology Co., Ltd. (Jiangmen, China) and authenticated by the Department of Pharmacognosy of Guangzhou Medical University (Voucher Specimen Number: GY-20220113). The 95% alcohol, n-butyl alcohol, phenol, antipyonin, methyl alcohol, trifluoroacetic acid (TFA), and deuterium oxide (D_2_O) were obtained from Macklin Biochemical Technology Co., Ltd. (Shanghai, China). Chloroform and concentrated sulfuric acid were sourced from Guangzhou Chemical Reagent Factory (Guangzhou, China). Monosaccharide standards (fucose, rhamnose, arabinose, galactose, glucose, xylose, mannose, fructose, ribose, galacturonic acid, glucuronic acid, mannuronic acid, and guluronic acid) and potassium bromide (KBr) were purchased from Beijing Solaibao Technology Co., Ltd. (Beijing, China). The BCA Protein Assay Kit was provided by GlpBio Biochemical Technology Co., Ltd. (Montclair, NJ, USA). The pureness for the remaining reagents was analytical pure.

### 2.2. Extraction and Purification of CRP

The peels were subjected to oven drying at 50 °C for 12 h prior to the extraction process. Dry peels were pulverized using a laboratory powder grinder and subsequently extracted with deionized water at a solid-to-liquid ratio of 1:15 (*w*/*v*) for three times at 95 °C for 90 min each. The extracts were allowed to cool to room temperature, and the supernatant was collected by a vacuum filtration device and subsequently concentrated at 50 °C with a rotary evaporator and precipitated with a final concentration of 95% alcohol at 4 °C for 12 h. Alcohol precipitation was collected following centrifugation at 5000 rpm for 15 min and subsequently redissolved in distilled water. The above-described alcohol precipitation procedure was performed in triplicate. Subsequently, deproteinization was conducted utilizing Selvage reagent (chloroform:butyl alcohol = 1:4, *v*/*v*). Finally, purified water-soluble polysaccharides were obtained by centrifugation and subsequently dialyzed against distilled water using a MD 3500 membrane (Millipore, Burlington, MA, USA) for 72 h. During the dialysis process, the distilled water was replaced every 4 h. The final purified polysaccharide (CRP) was obtained by lyophilizing at −80 °C. The yield of crude polysaccharides was calculated using the following equation: Yield (%) = (W1/W2) × 100%, where W1 represents the weight of the extracted polysaccharides (g), and W2 represents the weight of the original dry peels (g).

### 2.3. Structural Characterization of CRP

#### 2.3.1. Chemical Composition Identification

Further characterization of CRP was performed to determine its total carbohydrate and protein contents. The total carbohydrate content was quantified using the phenol-sulfuric acid method (5% phenol and 98% sulfuric acid). A standard calibration curve was established using glucose, and absorbance measurements were recorded at 490 nm with a UV-visible spectrophotometer. Protein concentration was quantified using the BCA assay kit. These systematic analyses collectively elucidated the physicochemical properties of CRP.

#### 2.3.2. Fourier-Transform Infrared (FT-IR) Analysis

An FT-IR spectrometer (Nicolet 6700, Thermo Fisher Scientific, Waltham, MA, USA) was employed to analyze the purified CRP via using the KBr tableting technique. CRP and KBr were weighed and ground into a fine powder at the ratio of 1: 100, and the mid-infrared region (4000–400 cm^−1^) was scanned in full-spectrum mode to acquire complete spectral profiles.

#### 2.3.3. Monosaccharide Composition Analysis

An appropriate amount of polysaccharide sample was accurately weighed and hydrolyzed with 1 mL of 2 M TFA at 121 °C for 2 h. After TFA-mediated hydrolysis, the mixture was then evaporated to dryness under a nitrogen stream, washed with 99.99% methanol, and re-dried. The hydrolyzed sample was subjected to iterative methanol washes (2–3 cycles) and resolubilized in sterile water prior to Ion chromatograph analysis (ICS 5000+, Thermo Fisher Scientific, USA). The separation utilized a Dionex™ CarboPac™ PA20 column (150 × 3.0 mm, 10 μm) under the following conditions: including mobile phases A (H_2_O), B (0.1 M NaOH), and C (0.1 M NaOH/0.2 M NaAc); at a flow rate of 0.5 mL/minute under 30 °C column temperature; and the injection volume was 5 μL. The gradient elution program was as follows: 0 min (95:5:0), 26 min (85:5:10), 42 min (85:5:10), 42.1 min (60:0:40), 52 min (60:40:0), 52.1 min (95:5:0), 60 min (95:5:0) (A/B/C, *v*/*v*). Fucose (Fuc), rhamnose (Rha), arabinose (Ara), galactose (Gal), glucose (Glc), xylose (Xyl), mannose (Man), fructose (Fru), ribose (Rib), galacturonic acid (GalA), glucuronic acid (GlcA), mannuronic acid (ManA), and guluronic acid (GulA) were used as monosaccharide standards.

#### 2.3.4. Nuclear Magnetic Resonance (NMR) Analysis

Accurately weigh 50 mg of vacuum-dried purified CRP sample and dissolve in 1.0 mL of D_2_O, followed by lyophilization. This deuterium exchange procedure was repeated twice. Subsequently, the lyophilized sample was redissolved in 0.5 mL of D_2_O. After ensuring complete dissolution, NMR spectra were characterized using a 600 MHz nuclear magnetic resonance spectrometer under low-temperature conditions, yielding both high-resolution ^1^H NMR and ^13^C NMR spectra. All NMR data were processed and analyzed using MestReNova software (version 14.0.0-23239).

### 2.4. Cell Culture

GES-1 cells were procured from GuangZhou Jennio Biotech (Guangzhou, China, JNO-H0240). The cells were cultured in RPMI 1640 medium (Gibco, New York, NY, USA) supplemented with 10% (*v*/*v*) heat-inactivated fetal bovine serum (FBS) and 1% penicillin-streptomycin solution (Gibco, USA), under an atmosphere of 5% CO_2_ at 37 °C. The cells were passaged regularly.

### 2.5. CCK-8 Assay

To select the optimal model of alcohol-induced GES-1 cell injury, GES-1 cells were seeded at a density of 5 × 10^3^ cells/well in 96-well plates and cultured for 24 h. The cells were then treated with various concentrations of ethanol (0, 0.2, 0.4, 0.6, 0.8, 1.0, 1.2 M) for 4 h. Subsequently, 10% CCK-8 reagent was added to each well and incubated for 2 h before measuring the optical density (OD) value. Cell viability was calculated to screen for the optimal alcohol-induced conditions.

For screening non-cytotoxic concentrations of CRP, GES-1 cells were seeded at 5 × 10^3^ cells/well in 96-well plates and cultured for 24 h. The cells were then treated with different concentrations of CRP (0.06, 0.2, 0.6, 1, 2, 4 mg/mL) for 24 h. Afterward, 10% CCK-8 reagent was added to each well and incubated for 2 h, followed by OD measurement. Cell viability was calculated to determine the optimal non-cytotoxic CRP concentration.

GES-1 cells in the logarithmic growth phase were seeded into 96-well plates at 5000 cells per well. Following 24 h incubation, cells were treated with CRP (1 mg/mL) for 24 h and then incubated with absolute alcohol (1 M) for 4 h. Subsequently, 10% CCK-8 solution (100 μL) was added to each well, followed by an additional incubation period of 2 h. Absorbance was measured at 450 nm using a multimode microplate reader (Epoch, Bio-Rad, Hercules, CA, USA). Then, the cell viability was calculated.

### 2.6. Immunofluorescence (IF) Staining

GES-1 cells in the logarithmic growth phase were seeded on glass coverslips in a 6-well plate (5000/well). Cells were pretreated with CRP (1 mg/mL) for 24 h and subsequently treated with alcohol (1 M) for 4 h. Afterward, the cells were washed with PBS, immersed with 4% paraformaldehyde (Servicebio, Shanghai, China) for 15 min, and permeabilized with 0.5% Triton X-100 (Beyotime, Shanghai, China) for 20 min at room temperature. Next, the cells were blocked with BSA (Jingxin, Guangzhou, China) for 60 min and then incubated overnight at 4 °C in the dark with primary antibodies against ZO-1, NLRP3, and Nrf2 (Affinity, Cincinnati, OH, USA). The following day, the cells were incubated with a rabbit anti-goat IgG (H + L) cross-adsorbed secondary antibody (Thermo Fisher, Waltham, MA, USA). Finally, the plates were sealed with anti-fluorescence quenching mounting medium containing DAPI (Beyotime, Shanghai, China) for at least 20 min. Fluorescence microscopy images of the sections were acquired using a Zeiss Axio Vert A1 microscope (Zeiss, Oberkochen, Germany).

### 2.7. Annexin V-FITC-Based Apoptosis Assay for Detecting Cell Apoptosis

GES-1 cells from the control group, alcohol-treated group, and CRP pretreatment group were harvested, resuspended in PBS, and mixed with 195 μL of loading buffer (Beyotime, Shanghai, China). Subsequently, 10 μL of PI and 5 μL of FITC were added, and the cells were stained at room temperature for 10 min [21,22]. The stained cells were then subjected to flow cytometry analysis, and the resulting data were processed using FlowJo software version 10.8.1 (FlowJo LLC, Ashland, OR, USA).

### 2.8. Wound Healing Assay

Prior to cell seeding, grid patterns consisting of five parallel lines (0.5 cm spacing) were inscribed on six-well culture plates using permanent markers to establish standardized observation fields. GES-1 cells were plated at 6 × 10⁵ cells/well and cultured until achieving 90% monolayer coverage. Mechanical wounds were then created perpendicular to the grid axis with an aseptic 200 μL pipette tip, followed by triple PBS washing to eliminate cellular debris. Next, the cells were pretreated with 1 mg/mL CRP for 24 h and then exposed to 1 M alcohol for 4 h. Images were taken using a Leica microscope (Leica, Wetzlar, Germany), and the results were analyzed quantitatively using ImageJ software (version V1.52).

### 2.9. Animal Grouping, Model Establishment, and Drug Administration

Thirty-two male Balb/c mice, aged 8 weeks and weighing approximately 40 ± 5 g, were purchased from the Guangdong Provincial Animal Center. Animals were group-housed by experimental treatment in cages at the Animal Experiment Center of Guangzhou Medical University. The environmental conditions were strictly controlled, with a temperature of 24–26 °C, humidity of 50–70%, and a 12 h light/dark cycle. Animals had an abundant supply of food and water. This study was approved by the Experimental Animal Ethics Committee of Guangzhou Medical University (Approval Number: GY2024-446).

The experimental design and the process for establishing an alcohol-induced gastric ulcer model are illustrated in Figure 1. After a one-week acclimation period to the new environment, mice were randomly assigned to four groups: the control group, the model group, the low-dose drug treatment group (50 mg/kg), and the high-dose drug treatment group (250 mg/kg), with eight mice in each group. The treatment groups were gavaged with CRP, whereas the control and model groups were gavaged with equal doses of saline. Both CRP and saline were administered once daily for five consecutive days. On the fifth day, all mice underwent a 16 h fasting period. In addition, 6 h after the last doses of either CRP or saline, the model and CRP groups were gavaged with edible liquor (56% alcohol, *v*/*v*, Red Star, Beijing, China), while the control group was gavaged with an equal volume of saline. Equal volumes of alcohol were administered at corresponding time points on days 6 and 7. On day 7, 4 h after gavage, blood samples were collected, and the mice were euthanized via cervical dislocation. The stomachs were promptly excised and longitudinally incised along the greater curvature, and sections of appropriate size were immersed in 4% paraformaldehyde (Beyotime, Shanghai, China). The remaining gastric tissues were immediately snap-frozen and stored at −80 °C for subsequent biochemical analysis.

### 2.10. Evaluation of Gastric Ulcer Degree

The degree and severity of the gastric ulcer in mice was evaluated by determining the ulcer area ratio. The Alpha 6400 Mirrorless camera (SONY, Tokyo, Japan) was employed to photograph the internal surfaces of stomachs. The gastric surface images in the photographs were selected using ImageJ software for quantification of the ulcer area. The ratio of gastric ulcers and the rate of ulcer inhibition were subsequently computed based on the total stomach area and the ulcer area. The gastric ulcer ratio in mice was calculated as the ulcer area on the inner surface of the stomach divided by the total gastric area of the inner surface.

### 2.11. Western Blot Analysis

Mice gastric tissues or GES-1 cells were subjected to homogenization (on ice) using RIPA lysis buffer (Beyotime, Shanghai, China). The protein content assessment was carried out with BCA assay kits (Thermo Fisher, Waltham, MA, USA). Equal amounts of protein samples (30 μg) were separated by 10% SDS-PAGE and subsequently transferred onto PVDF membranes. The membranes were blocked with 5% skim milk and incubated overnight at 4 °C with specific primary antibodies. The antibodies used include β-actin, GAPDH, β-tubulin, TNF-α, iNOS, COX-2, NLRP3, HO-1, Nrf2, ZO-1, Claudin-5, and occludin from Proteintech (Wuhan, China), and cleaved caspase-1 and ASC from Affinity (Cincinnati, USA). Subsequently, horseradish peroxidase-conjugated secondary antibodies were incubated with the membranes for 1 h at room temperature. Protein band intensities were detected using an enhanced chemiluminescence (ECL) reagent and quantified using ImageJ software.

### 2.12. Assessment of the Oxidative Stress Index

Fresh blood was rapidly collected prior to mouse sacrifice and centrifuged at 5000 rpm for 30 min at 4 °C. The resulting supernatant and harvested gastric tissues were stored at −80 °C. Superoxide dismutase (SOD) activity and malondialdehyde (MDA) content were determined following the manufacturer’s protocols (Nanjing Jiancheng Bioengineering Institute, Nanjing, China). Reactive oxygen species (ROS) production induced by CRP and alcohol treatment in GES-1 cells and gastric tissues was assessed by measuring the fluorescence intensity of 2′,7′-dichlorofluorescein diacetate (DCFH-DA) probes (Abcam, Cambridge, UK).

### 2.13. Hematoxylin and Eosin (H&E) Staining

The fixed gastric tissues were sequentially dehydrated, embedded in paraffin, and subsequently sectioned into uniform slices. After removing the paraffin, the sections were stained with hematoxylin (6 min) and eosin (1 min) to facilitate histopathological examination. The pathological alterations in the gastric tissues of mice from each group were systematically evaluated under an optical microscope (Leica, Wetzlar, Germany). The gastrointestinal inflammation and gastric mucosal injury grading in this article is based on the following reference article [23,24], with the specific criteria and scores provided in Appendix A.

### 2.14. Immunohistochemistry (IHC) Staining

Gastric tissue specimens were fixed in 4% paraformaldehyde for 48 h and processed through standard dehydration, embedding, and sectioning protocols. Following deparaffinization with xylene and rehydration through a graded ethanol series, antigen retrieval was performed. Endogenous peroxidase activity was blocked with 3% H_2_O_2_ (10 min). Sections were incubated with 5% goat serum (30 min) before overnight incubation at 4 °C with primary antibodies (iNOS, Nrf2, NLRP3, ZO-1) diluted in antibody diluent. After PBS washes and room temperature equilibration, biotin-conjugated secondary antibodies were applied (45 min), followed by HRP-streptavidin incubation (30 min). Signal development was achieved using DAB substrate under microscopic monitoring, with hematoxylin counterstaining. Sections were dehydrated, cleared, and mounted with neutral resin for microscopic analysis. All antibodies were diluted in PBS containing 5% goat serum.

### 2.15. qRT-PCR Analysis

After administration and modeling, total RNA was extracted from GES-1 cells using TRIzol reagent and converted into cDNA using the Prime Script RT Reagent Kit. qRT-PCR was performed using specific primers and SYBR Select master mix, with amplification monitored on a 7300 Real-Time PCR System (Applied Biosystems, Foster City, CA, USA). The amplification of target cDNA was facilitated by the primers listed in Appendix A, following 40 cycles at 95 °C for 30 s, primer annealing at 95 °C for 5 s, and extension at 60 °C for 30 s. mRNA levels were quantified relative to the housekeeping gene GAPDH using the 2 −ΔΔCt method. All primers were validated, and relative expression levels were compared to those of the control group.

### 2.16. Statistical Analysis

The experimental data were statistically analyzed using GraphPad Prism 10 (GraphPad Software, Inc., San Diego, CA, USA). The results are expressed as the mean ± standard deviation (SD). Differences between groups were analyzed by one-way ANOVA with Tukey’s post hoc test; statistical significance was defined as * *p* < 0.05.

## 3. Results

### 3.1. Extraction, Purification, and Preliminary Characterization of CRP

A pectic polysaccharide fraction was obtained from *Citrus reticulata* ‘Chachi’ via aqueous extraction followed by alcohol precipitation, deproteinization, dialysis, and lyophilization. The yield of purified pectic polysaccharide was 4.28% based on the dry weight of the alcohol insoluble solid (AIS) [25]. The total carbohydrate contents of CRP were 69.78%. The BCA method showed that the protein concentration of it was 2.03%. Monosaccharide composition analysis of CRP revealed that CRP was predominantly composed of Ara (49.68%), GalA (26.02%), Gal (10.70%), Rha (5.25%), Glc (4.34%), Man (3.26%), and Xyl (0.73%) (Figure 2A). FT-IR spectroscopy revealed characteristic absorption bands corresponding to the polysaccharide backbone (Figure 2B). The strong absorption peak at 3430.44 cm^−1^, characterized by its broad shape, can be ascribed to O-H stretching vibrations. Similarly, the band observed at 2948.92 cm^−1^ is associated with C-H stretching vibrations. The stretching vibrations of esterified carboxylic acid groups (-COOR) associated with the absorption peaks at 1745.20 cm^−1^ and the free carboxylic acid group (-COO^−^) were showed in the band at 1625.70 cm^−1^, which was caused by the carboxylic acid group of galacturonic acid that exists in two distinct forms in CRP. Additional peaks at 1425.94 cm^−1^ and 1346.27 cm^−1^ corresponded to C-H bending vibrations, and 1237.34 cm^−1^ was attributed to the C-O stretching vibration within the carboxylic acid group. The strong band at 1146.20 cm^−1^ and 1103.10 cm^−1^ was associated with glycosidic linkage C-O-C stretching vibrations. The characteristic peak for α-glycosidic and β-glycosidic bonds was observed at 966.19 cm^−1^ and 815.07 cm^−1^. Furthermore, the absence of discernible absorption bands in the Amide I (1658–1650 cm⁻^1^) regions indicates negligible protein content within the sample. The situation also exhibited a similar pattern in the Amide II region (1555–1530 cm⁻^1^). ^1^H NMR and ^13^C NMR spectroscopy revealed the fine structures of CRP (Figure 2C,D). On the spectrogram of ^1^H NMR, the anomeric proton signals of H-1 were at 5.18 ppm, 5.10 ppm, 5.08 ppm, 5.04 ppm, 5.03 ppm, 5.01 ppm, 4.89 ppm, 4.83 ppm, 4.75 ppm, 4.69 ppm, 4.57 ppm, and 4.56 ppm. The peak near 3.73 ppm was attributed to carboxyl methyl esterification of GalA units, with pronounced peak intensity, which seems to imply a high abundance of methylesterified GalA residues in CRP. In addition, the results of ^13^C NMR spectroscopy show that the signals at 171.23 ppm, 170.71 ppm, and 170.24 ppm were assigned to the ester carbonyl carbons at the C-6 position of methyl-esterified GalA, while the signal at 174.33 ppm was attributed to the free carboxyl carbon at the C-6 position of non-esterified GalpA, which means methyl-esterified galacturonic acid predominated over its non-esterified counterpart. This observation is consistent with the results obtained from infrared spectroscopy. Based on the characteristic resonances observed in the anomeric region of the ^13^C NMR, the corresponding anomeric carbon signals of C-1 were identified at 107.50 ppm, 107.43 ppm, 107.15 ppm, 107.08 ppm, 106.91 ppm, 106.86 ppm, 106.80 ppm, 106.50 ppm, 104.35 ppm, 100.40 ppm, 100.02 ppm, and 99.66 ppm. Meanwhile, the peak at 52.85 ppm was attributed to carboxyl methyl esterification of GalA. Moreover, the signals at 1.23 ppm (^1^H NMR) and 16.55 ppm (^13^C NMR) were assigned to the characteristic methyl protons of Rha.

### 3.2. CRP Protected Against Alcohol-Induced Loss of Body Weight and Gastric Mucosal Damage in BALB/c Mice

During the medication pretreatment, the average weight of mice in each group remained consistently stable throughout the observation period. Subsequent to alcohol-induced modeling, we noted a substantial decrease in body weight among the alcohol-induced cohort. Conversely, both CRP-pretreated groups showed a significant enhancement in body weight relative to the alcohol-induced group, with the 250 mg/kg group exhibiting body weight similar to that of the control group (Figure 3A).

Macroscopic observation revealed that the pretreated groups receiving CRP at doses of 50 and 250 mg/kg exhibited a significant reduction in gastric lesions compared to the alcohol-induced group, where alcohol induced intense gastric mucosal damage that appeared in the form of hemorrhages. Furthermore, almost no distinct hemorrhagic lesions could be seen on gastric tissues of the high-dose CRP group (Figure 3C and Appendix A).

Histopathological alterations in stomach sections from the different groups are presented in Figure 3D and Appendix A. No histological abnormalities were detected in the control group, whereas the model group exhibited marked damage and extensive edema, with reduced glandular structures and infiltration of inflammatory cells into the mucosal layer (As indicated by the arrow). In contrast, CRP-pretreated groups were significantly improved, exhibiting less mucosal damage and milder edema compared to the model group.

### 3.3. CRP Alleviated Gastric Inflammatory Damages in Alcohol-Induced GU Mice

The alcohol group significantly elevated the protein expression levels of pro-inflammatory cytokines TNF-α and IL-1β in the gastric tissue of rats. Additionally, the expression levels of the inflammatory regulators iNOS and COX-2 were elevated by 1.78-fold and 1.23-fold compared to those in the normal group, respectively. In this study, CRP pretreatment markedly counteracted the alcohol-induced upregulation of these inflammatory markers. Notably, the expression levels of TNF-α and IL-1β in the high-dose CRP pretreatment group were comparable to those in the control group while further suppressing iNOS and COX-2 expression below baseline control values (Figure 4A). Furthermore, IHC was performed to evaluate the expression patterns of iNOS (Figure 4C). Compared to the normal group, the alcohol-induced group showed a markedly increased expression of iNOS, indicating significant upregulation. In contrast, pretreatment with CRP resulted in a concentration-dependent downregulation of iNOS expression, with marked suppression observed at higher doses (the IHC magnification is shown in Appendix A).

### 3.4. CRP Inhibited NLRP3 Inflammasome Activation in Alcohol-Induced GU Mice

The robust release of TNF-α and IL-1β is closely linked to the activation of the NLRP3/ASC/Caspase-1 signaling pathway [26,27]. To further investigate the anti-inflammatory effects of CRP, we analyzed the expression levels of key proteins in this pathway, including NLRP3, ASC, and cleaved caspase-1. Notably, alcohol exposure significantly upregulated NLRP3, ASC, and cleaved caspase-1 expression in the alcohol-induced group, whereas CRP pretreatment markedly suppressed these elevations. Furthermore, CRP dose-dependently inhibited the expression of all three proteins (Figure 5A). Subsequently, IHC was performed to assess NLRP3 inflammasome expression in gastric tissues across experimental groups. The results demonstrated a marked upregulation of NLRP3 in the alcohol-induced group compared to control, with pronounced immunostaining localized to mucosal epithelial cells (Figure 5B) (the IHC magnification is shown in Appendix A). Strikingly, pretreatment with CRP significantly attenuated NLRP3 expression, suggesting suppression of inflammasome activation. Collectively, these findings indicate that CRP pretreatment effectively reverses alcohol-triggered hyperactivation of inflammatory pathways through NLRP3-dependent mechanisms, specifically, by attenuating the NLRP3/ASC/Caspase-1 signaling axis.

### 3.5. CRP Decreased Lipid Peroxidation Level, SOD Activity, and the Production of ROS in Alcohol-Induced GU Mice

The results demonstrate that SOD activity was reduced in the alcohol-induced group compared to the control group, likely attributable to oxidative stress induced by alcohol exposure. However, animals pretreated with CRP exhibited markedly higher SOD activity compared to the ulcer control group. These findings indicate the potential role of CRP’s cellular antioxidant activity in safeguarding the gastric mucosa against alcohol-induced damage in the animal ulcer model (Figure 6A).

Results indicate that the MDA level in intact gastric mucosa was detected at an extremely low level, while a significantly higher MDA level was observed in the alcohol-induced group. This increase is attributed to lipid peroxidation and oxidative damage caused by alcohol administration. However, animals pretreated with CRP at doses of 50 and 250 mg/kg demonstrated statistically significant reductions in MDA levels compared to the alcohol-induced group. These findings suggest the potential protective ability of CRP against alcohol-induced lipid peroxidation in the gastric mucosa (Figure 6B).

To detect the effects of CRP pretreatment on ROS levels, gastric tissues from each group were enzymatically digested into single-cell suspensions for flow cytometric analysis. Notably, our findings demonstrated a marked elevation of ROS levels in the alcohol-challenged group compared to controls. In contrast, CRP-pretreated groups exhibited an obvious attenuation of ROS generation, with levels restored to near-baseline values (slightly exceeding control levels) (Figure 6C,D).

### 3.6. CRP Improved Gastric Mucosal Oxidative Damage by Activating Nrf2 Signaling Pathway

To investigate the antioxidant mechanism of CRP in alcohol-induced GU, we examined the expression levels of Nrf2 and HO-1 using Western blot analysis (Figure 7A). Compared to the control group, the alcohol-induced group exhibited significant downregulation of both Nrf2 and HO-1. Strikingly, CRP pretreatment (250 mg/kg) reversed this suppression, upregulating Nrf2 and HO-1 expression to 2.1 ± 0.8-fold and 1.6 ± 0.5-fold, respectively, and restoring protein levels to near-baseline values (112 ± 12% and 86 ± 9% of control).

IHC analysis further corroborated these findings, demonstrating pronounced Nrf2 downregulation in the alcohol-induced group, which was markedly rescued by CRP pretreatment (Figure 7B) (the IHC magnification is shown in Appendix A).

Collectively, these data demonstrate that pretreatment with CRP alleviates alcohol-induced oxidative stress and mucosal injury through activation of the Nrf2/HO-1 signaling pathway.

### 3.7. CRP Stimulated the Expression of ZO-1, Occludin, and Claudin5 in the Gastric Mucosa of Alcohol-Induced Mice

Western blot analysis demonstrated that alcohol exposure significantly downregulated the expression levels of ZO-1, Occludin, and Claudin5 in the gastric mucosa of mice. Strikingly, CRP pretreatment (250 mg/kg) restored these proteins to near-physiological levels (Figure 8A). IHC analysis corroborated these findings, demonstrating intense ZO-1 immunoreactivity (brown staining) in control mucosal epithelial cells, whereas alcohol-induced lesions exhibited near-complete ZO-1 suppression (Figure 8B) (the IHC magnification is shown in Appendix A).

Collectively, CRP mitigates alcohol-induced mucosal barrier dysfunction by preserving tight junction architecture through coordinated upregulation of ZO-1, occludin, and Claudin-5.

### 3.8. CRP Improves Survival in Alcohol-Induced GES-1 Cells Injury

GES-1 cells were treated to 1.0 M alcohol for 4 h to establish the damage model (Figure 9A). The cell viability of GES-1 cells treated with various concentrations of CRP for 24 h is shown in Figure 9B. The optimal dose of CRP was determined to be 1 mg/mL (Figure 9C). FITC-PI staining analysis demonstrated that the cell viability in the alcohol group was reduced compared to the control (Figure 9D). In contrast, cell viability was markedly enhanced in the groups pretreated with CRP (Figure 9D). Also, wound healing capacity analysis illustrated the findings of our investigation into the protective role of CRP pretreatment in mitigating alcohol-induced cell damage (Appendix A).

### 3.9. CRP Inhibited the NLRP3 Inflammasome Activation of GES-1 Cells

Alcohol stimulation upregulated the inflammatory protein levels of NLRP3, ACS, and cleaved caspase-1. Interestingly, CRP pretreatment decreased the levels of Alcohol-stimulated NLRP3 (30.42 ± 20.70%), ACS (51.26 ± 23.55%), and cleaved caspase-1 (64.1 ± 20.10%) (Figure 10). Furthermore, IF staining experiments revealed that the expression of NLRP3 was upregulated in the alcohol group compared with the control group. However, CRP pretreatment effectively attenuated this alcohol-induced increase in NLRP3 levels (Appendix A).

### 3.10. CRP Ameliorates Alcohol-Induced Oxidative Stress in GES-1 Cells Through Modulating Nrf2/HO-1 Pathway

The impact of CRP on intracellular ROS production in GES-1 cells was assessed by measuring the fluorescence intensity of DCF, which is generated through the oxidation of the fluorogenic probe DCFH-DA. Flow cytometric analysis demonstrated that CRP pretreatment significantly attenuates the alcohol-induced substantial increase in ROS production (Figure 11A).

Next, we investigated the molecular mechanisms by which CRP protects against alcohol-induced GES-1 cell injury. Using Western blot analysis, we noticed that Nrf2 and HO-1 protein levels were significantly reduced in alcohol-exposed GES-1 cells compared to untreated cells. However, pretreatment with CRP restored the levels of Nrf2 and HO-1 in alcohol-treated GES-1 cells (Figure 11B). Similar to these findings, the fluorescence intensity of Nrf2 exhibited a consistent trend via immunofluorescence analysis, and qRT-PCR results also demonstrated a parallel trend (Appendix A).

The in vitro results mirrored the in vivo outcomes, reinforcing the mechanistic consistency across experimental models.

## 4. Discussion

In this study, we identified that CRP primarily consists of seven monosaccharide components: Ara, GalA, Gal, Rha, Glc, Man, and Xyl. Notably, Ara and Gal emerged as the dominant neutral monosaccharides, collectively accounting for 60.38% of the total monosaccharide composition. Among neutral monosaccharides, Ara exhibited the highest content in CRP, followed by Gal. Previous investigations have established the therapeutic significance of Ara across diverse disease models, including metabolic syndrome, inflammatory disorders, food allergy, and oxidative stress mitigation [28,29]. A comprehensive review by Csaba Fehér identified L-arabinose as a hypoglycemic agent that exhibits protective effects against hyperglycemia and demonstrates significant antioxidant properties [30]. Furthermore, arabinose can be enzymatically reduced to arabinitol, a sugar alcohol that has been shown to exhibit anti-inflammatory properties [31]. Concurrently, Gal was observed to play a pivotal role in antioxidant activities [32], with accumulating in vitro and in vivo evidence confirming CRP’s substantial antioxidant and anti-inflammatory effects. In contrast, GalA, as an acidic monosaccharide, displayed comparatively lower antioxidant activity than its neutral monosaccharides [33]. However, it may function as an efficient free radical scavenger [34], potentially contributing to CRP’s anti-gastric ulcer effects through neutralization of reactive oxygen species generated during alcohol intoxication. This mechanistic distinction highlights the multifaceted bioactivity profile of CRP’s monosaccharide constituents, with neutral and acidic components synergistically mediating its pharmacological actions through complementary pathways.

Importantly, the structural characterization, particularly the high GalA and Ara content, strongly indicates that CRP possesses properties characteristic of the pectic polysaccharide family. Accumulating evidence suggests that pectic polysaccharides can serve as fermentable substrates for the gut microbiota, being metabolized into bioactive oligosaccharides and short-chain fatty acids such as acetate, propionate, and butyrate [35]. These microbial metabolites possess well-documented systemic anti-inflammatory properties [36]. Given the observed potent systemic anti-inflammatory effects of CRP in our alcohol-induced ulcer model, it is plausible that microbial transformation of CRP in the lower gastrointestinal tract contributes to its overall gastroprotective efficacy. While our current study focused on elucidating the direct mechanisms of CRP within the gastric mucosa using acute models, investigating the potential role of gut microbiota in CRP’s metabolism and its contribution to the observed protective effects represents a crucial and exciting direction for future translational research.

The alcohol-induced GU model in mice is a well-established paradigm for evaluating antiulcer agents [37]. Multiple studies have confirmed that alcohol rapidly infiltrates the gastric mucosa, triggering oxidative stress and inflammatory cascades, resulting in epithelial denudation, submucosal edema, and hemorrhage [38]. In our study, CRP pretreatment (50/250 mg/kg) dose-dependently attenuated alcohol-induced mucosal lesions, with histological analysis revealing preserved mucosal architecture and reduced leukocyte infiltration.

Alcohol-induced inflammatory responses can activate the innate immune system, leading to altered levels of inflammatory cytokines [39,40]. Pro-inflammatory cytokines, particularly TNF-α, are key mediators in the development of GU inflammation and tissue damage [41]. Previous studies have consistently shown that elevated TNF-α levels are closely associated with inflammatory diseases, such as mucosal inflammation, and can impair gastric microcirculation, ultimately leading to delayed healing of GU [42,43,44,45]. Furthermore, TNF-α can also regulate the expression of other inflammatory molecules, including iNOS, COX-2, and IL-1β [46]. Interleukins have been shown to possess the ability to suppress acute inflammatory responses and modulate mucosal defense barriers [47,48]. In this study, we found that CRP pretreatment significantly suppressed the levels of TNF-α, IL-1β, iNOS, and COX-2, thereby exerting potent anti-inflammatory effects (Appendix A).

The NLRP3 inflammasome plays a crucial and multifaceted role in the human immune response, serving as a critical mediator of inflammatory responses and programmed cell death under pathological or stress conditions [49]. This molecular complex, composed of NLRP3, ASC, and Caspase-1, functions as a pattern recognition receptor that activates downstream effector Caspase-1 to regulate chronic inflammation [50]. Regarding in vivo regulation, CRP pretreatment (250 mg/kg) significantly reversed alcohol-induced upregulation of NLRP3, ASC, and Cleaved-caspase1 in mice gastric tissues. Parallel experiments in alcohol-treated GES-1 cells showed CRP similarly suppressed NLRP3 inflammasome components. These consistent in vivo and in vitro results establish that CRP exerts gastroprotective effects through targeted modulation of the NLRP3/ASC/Caspase-1 signaling axis.

Numerous studies have extensively investigated the critical role of oxidative stress in the pathogenesis of alcohol-induced gastric injury [51], which is closely associated with alterations in ROS, antioxidant enzymes, and oxidative byproducts [52]. Alcohol stimulation triggers excessive ROS production [53,54], disrupting the homeostasis between pro-oxidants and the antioxidant defense system [55]. As the primary frontline defense against ROS, SOD is renowned for its capacity to scavenge superoxide radicals [56,57]. The observed surge in MDA levels results from ROS-mediated membrane lipid peroxidation, which subsequently exacerbates oxidative stress damage [58,59]. Substantial evidence confirms that compounds with antioxidant properties can effectively mitigate alcohol-induced gastric injury [60]. In our current investigation, alcohol treatment significantly upregulated ROS expression in vivo, whereas CRP pretreatment demonstrated remarkable ROS suppression. Parallel experiments in GES-1 cells yielded consistent results. Furthermore, alcohol exposure markedly decreased SOD activity while increasing MDA levels in gastric tissues—pathological changes that were effectively reversed by CRP pretreatment. These findings collectively demonstrate CRP’s potent antioxidative capacity in counteracting alcohol-induced oxidative damage.

Nrf2 is a well-known target for regulating redox balance, and it exerts its effects by promoting the production of antioxidant enzymes [61]. Under oxidative stress, Nrf2 undergoes Keap1-mediated derepression, subsequently migrating to the nucleus, where it engages with ARE sequences to initiate transcriptional activation of cytoprotective enzymes including HO-1, NQO1, and SOD2 [62,63]. Among these, the HO-1 gene regulated by Nrf2 has the closest relationship with gastrointestinal protection [64]. The downstream products of HO-1 are involved in regulating various physiological processes, such as anti-inflammation, anti-apoptosis, antioxidation, and immunomodulation, demonstrating that the induction of HO-1 may alleviate the inflammatory response [65]. The Nrf2/HO-1 signaling pathway plays a pivotal role in protecting the gastric mucosa against injury [66]. The two exert crucial protective effects through the interactive dialogue that regulates antioxidant events in gastric injury. Our experimental data demonstrated that alcohol exposure appreciably decreased the expression of Nrf2 and HO-1 in gastric tissues. This inhibition was correlated with increased levels of the oxidative damage marker malondialdehyde (MDA) and decreased antioxidant capacity, as indicated by reduced superoxide dismutase (SOD) activity. However, CRP pretreatment effectively reversed the alcohol-induced suppression of Nrf2 and restored HO-1 expression to near-control levels. Parallel experiments in alcohol-stimulated GES-1 cells further confirmed that CRP could dose-dependently activate Nrf2 and synergistically upregulate HO-1. These in vivo and in vitro results collectively demonstrate that CRP pretreatment induces gastric protection via antioxidative stress mechanisms by activating the Nrf2/HO-1 pathway, along with the modulation of SOD activity and MDA levels.

The gastric mucosal barrier serves as the host’s primary line of defense against mucosal injury [67]. This barrier comprises three critical components: (1) a tightly packed epithelial cell layer, (2) a specialized mucus coating, and (3) bicarbonate ions [68]. TJs—localized between gastric epithelial cells—play a pivotal role in maintaining epithelial integrity by forming impermeable seals that separate physiologically distinct compartments and block luminal antigens [69]. Consequently, TJs proteins are key determinants of gastric mucosal homeostasis. Notably, apoptosis of gastric mucosal epithelial cells directly leads to disruption of tight junction structures, subsequently triggering inflammation and ulcer formation [70]. In this study, we conducted CCK-8 assays, Annexin-V/PI dual staining experiments, and scratch tests, revealing that CRP pretreatment markedly enhanced the viability of GES-1 cells. Furthermore, marked upregulation of tight junction proteins Claudin-5, Occludin, and ZO-1 was detected following CRP intervention (Appendix A). These results collectively indicate that CRP plays a critical and protective role in the gastric mucosal defense mechanisms against injury, potentially by reinforcing epithelial barrier integrity and enhancing cellular repair capacity.

While our findings robustly demonstrate the association of CRP’s gastroprotection with dual modulation of the NLRP3/ASC/Caspase-1 and Nrf2/HO-1 pathways, it is crucial to acknowledge that the mechanistic evidence presented remains correlational. To unequivocally establish causal relationships and strengthen the mechanistic claims, future studies should employ targeted approaches, such as pharmacological inhibition (e.g., using Nrf2 inhibitors like ML385 or NLRP3 blockers like MCC950) or genetic silencing (e.g., siRNA knockdown of Nrf2 or NLRP3). These experiments would clarify whether the observed anti-inflammatory and antioxidant effects are directly dependent on these specific pathways. This study also has several translational limitations that warrant attention. First, the bioavailability of CRP remains unclear, which may impact its clinical application potential. Additionally, the toxicity profile of CRP—including potential adverse effects and drug–herb interactions—was not evaluated. Future research should include pharmacokinetic studies to assess CRP’s in vivo absorption, distribution, metabolism, and excretion, as well as safety/toxicity testing via acute and chronic toxicity experiments in animal models. Moreover, the current focus on alcohol-induced gastric ulcers limits the generalizability of our findings. CRP’s protective effects against ulcers induced by other etiologies, such as H. pylori infection or NSAIDs, require validation using relevant animal models. Finally, the lack of data on potential interactions between CRP and conventional ulcer medications (e.g., proton pump inhibitors) hinders clinical translation, necessitating future drug combination studies. Additionally, it is crucial to clarify that the alcohol-induced mouse model employed in this study represents acute gastric mucosal injury, not chronic ulceration. This model elicits rapid pathological changes, including congestion, edema, hemorrhage, and superficial epithelial necrosis/erosion (as depicted in Figure 3C), following short-term exposure to high-concentration alcohol. Future studies are warranted to evaluate the long-term efficacy and therapeutic potential of CRP in more relevant chronic ulcer models, such as those induced by *Helicobacter pylori* infection [71]. Nonetheless, the consistent in vivo and in vitro data provide a compelling foundation for addressing these limitations and advancing CRP’s translational potential.

## 5. Conclusions

This study systematically elucidates the gastroprotective mechanisms of CRP against alcohol-induced GU through integrated in vivo and in vitro investigations. CRP pretreatment demonstrates multifaceted efficacy by robustly suppressing alcohol-triggered inflammatory cascades, notably, downregulating pro-inflammatory cytokines (TNF-α, IL-1β) and mediators (iNOS, COX-2) while inhibiting NLRP3 inflammasome hyperactivation via reduced expression of NLRP3, ASC, and cleaved caspase-1. Concurrently, CRP mitigates oxidative stress by activating the Nrf2/HO-1 antioxidant axis, effectively neutralizing ROS overproduction and lipid peroxidation (evidenced by restored SOD activity and reduced MDA levels), thereby preserving redox homeostasis. Furthermore, CRPs upregulate TJs proteins expression (occludin, ZO-1, Claudin-5) to restore both the structural and functional integrity of the gastric mucosal barrier, consequently preventing ulcer progression. Notably, this work represents the first mechanistic investigation focusing on the polysaccharide fraction of *Citrus reticulata*, revealing its dual protective functions encompassing both physical barrier restoration and bioactive regulation. These findings not only provide modern scientific validation for the traditional medicinal applications of *Citrus reticulata* but also provide a theoretical foundation for developing natural product-based multi-target strategies against ulcer pathogenesis.

## Figures and Tables

**Figure 1 nutrients-17-02062-f001:**
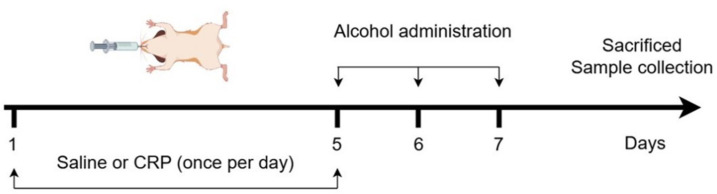
Experimental design and process of establishing alcohol-induced GU mice. Thirty-two male BALB/c mice (8 weeks of age) were randomly assigned to 4 experimental groups. Control group: Daily intragastric administration of saline. Model group: Daily saline administration for the first 5 d followed by alcohol exposure. Low-dose CRP group: Daily 50 mg/kg CRP administration with alcohol exposure. High-dose CRP group: Daily 250 mg/kg CRP administration with alcohol exposure.

**Figure 2 nutrients-17-02062-f002:**
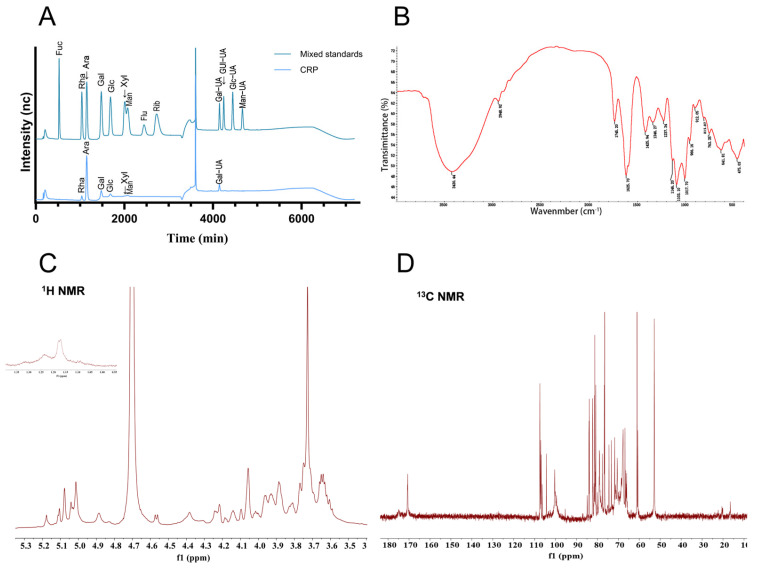
Characterization of *Citrus reticulata* ‘Chachi’ polysaccharides (CRPs). (**A**) Monosaccharide composition of CRP. (**B**) Fourier-transform infrared (FT-IR) spectrum of CRP. (**C**,**D**) ^1^H NMR spectrum and ^13^C NMR spectrum of CRP.

**Figure 3 nutrients-17-02062-f003:**
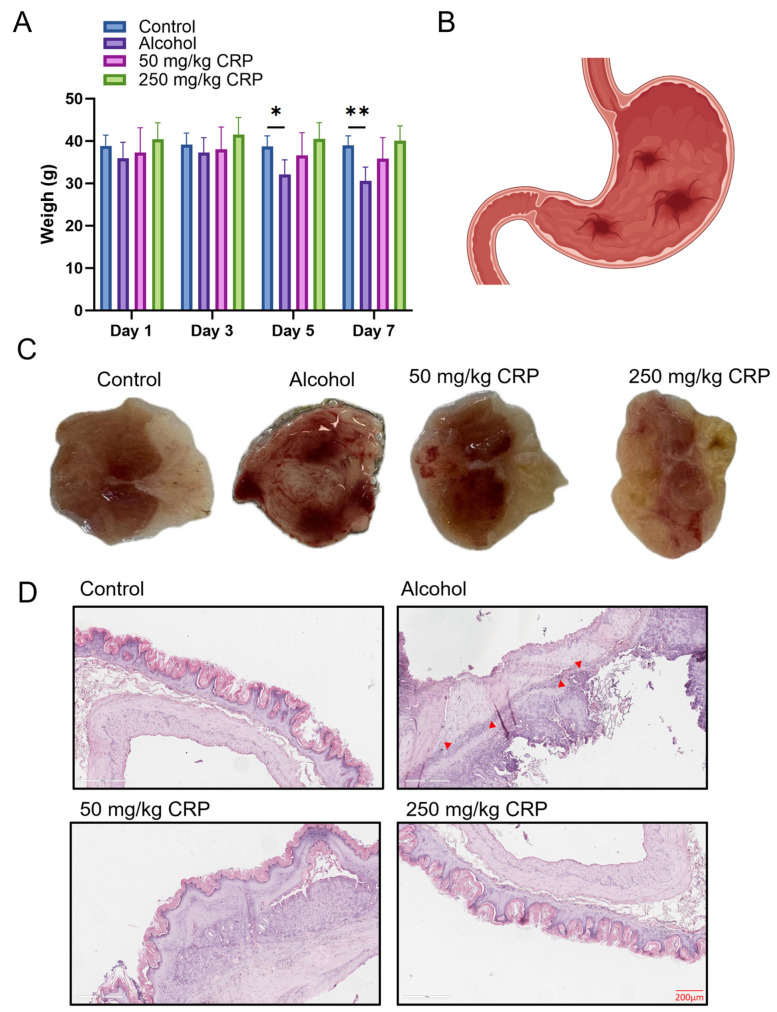
Effect of CRP on body weight and gastric damage in alcohol-induced GU mice. (**A**) Changes in body weight. (**B**) Schematic representation of gastric tissue structure. (**C**) Typical image of gastric mucosa. (**D**) H&E staining (scale bar: 200 μm). The arrows indicated damage and edema, reduced glandular structures and infiltration of inflammatory cells. Data are presented as means ± SD (n = 8). * *p* < 0.05, ** *p* < 0.01 versus the alcohol-induced group.

**Figure 4 nutrients-17-02062-f004:**
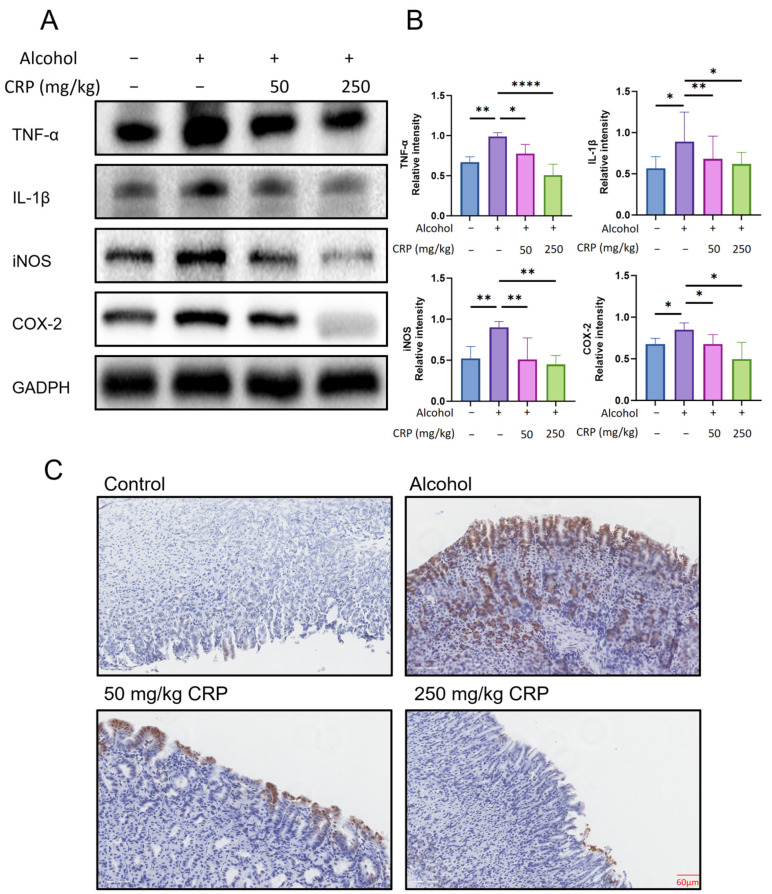
Effect of CRP on inflammatory markers in alcohol-induced GU mice. (**A**,**B**) Western blot detects the expression of inflammatory markers, including TNF-α, IL-1β, iNOS, and COX-2. GAPDH was utilized as a loading control to ensure equal protein loading. Data are presented as means ± SD (n = 8). * *p* < 0.05, ** *p* < 0.01, **** *p* < 0.0001 versus the alcohol-induced group. (**C**) IHC of iNOS enzyme expression in the gastric mucosa (scale bar: 60 μm).

**Figure 5 nutrients-17-02062-f005:**
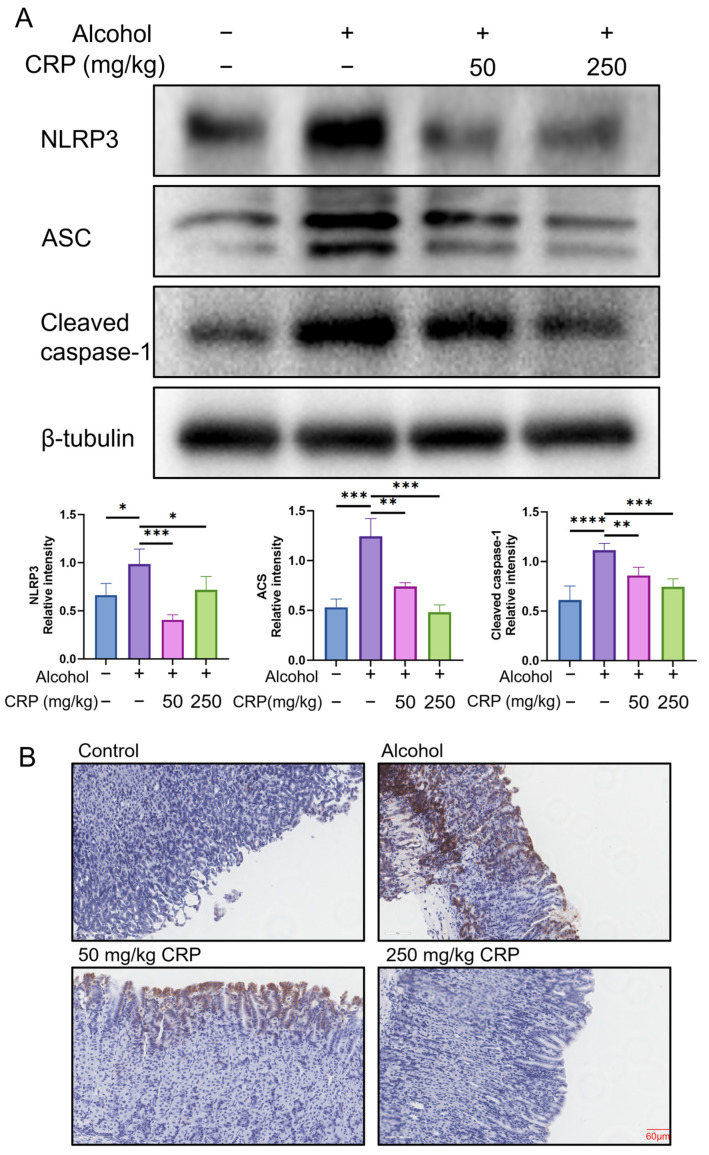
Effect of CRP pretreatment on alcohol-induced activation of NLRP3 inflammasome in GU mice. (**A**) Western blot detects the crucial protein expressions of the NLRP3/ASC/Caspase-1 signaling pathway. β-tubulin was used as a loading control to ensure equal protein loading. Data are presented as means ± SD (n = 8). * *p* < 0.05, ** *p* < 0.01, *** *p* < 0.001, **** *p* < 0.0001 versus the alcohol-induced group. (**B**) IHC of NRLP3 expression in the gastric mucosa (scale bar: 60 μm).

**Figure 6 nutrients-17-02062-f006:**
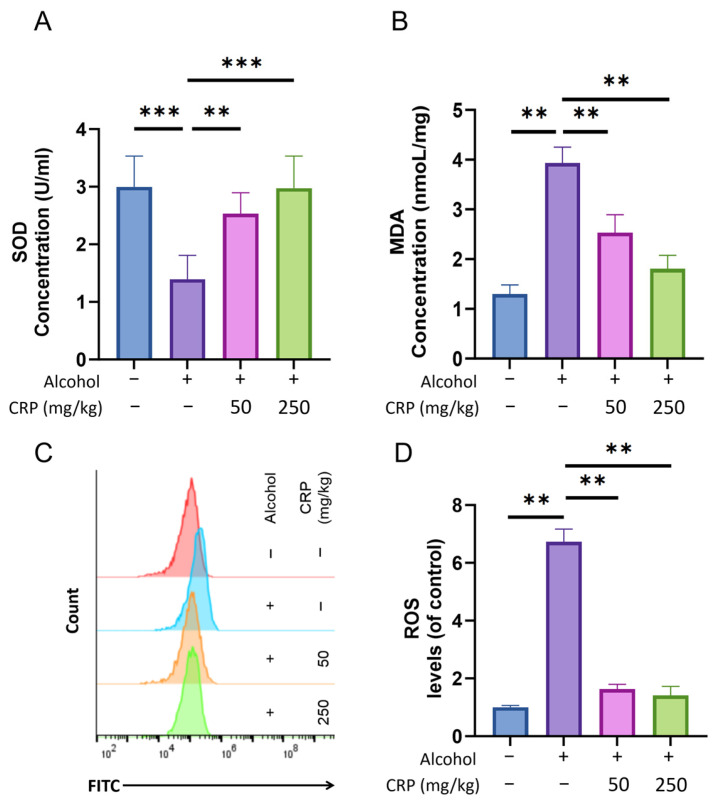
Effect of CRP pretreatment on oxidative stress markers in alcohol-induced GU Mice. (**A**) SOD levels in gastric tissues. (**B**) MDA levels in gastric tissues. (**C**,**D**) ROS levels quantified by flow cytometry. Data are presented as means ± SD (n = 8). ** *p* < 0.01, *** *p* < 0.001 versus the alcohol-induced group.

**Figure 7 nutrients-17-02062-f007:**
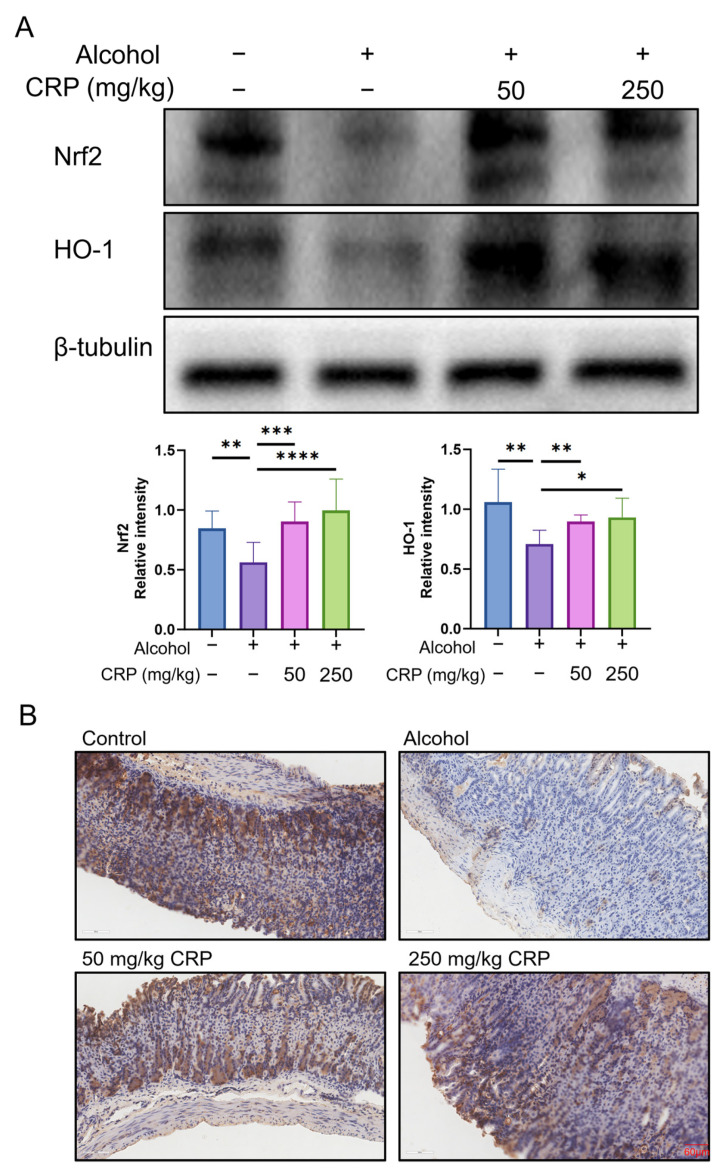
Effect of CRP pretreatment on alleviating alcohol-induced gastric oxidative damage through Nrf2 pathway activation in mice. (**A**) The expression levels of Nrf2 and HO-1 in gastric tissues were analyzed by Western Blot. β-tubulin was used as a loading control to ensure equal protein loading. Data are presented as means ± SD (n = 8). * *p* < 0.05, ** *p* < 0.01, *** *p* < 0.001, **** *p* < 0.0001 versus the alcohol-induced group. (**B**) IHC of Nrf2 expression in the gastric mucosa (scale bar: 60 μm).

**Figure 8 nutrients-17-02062-f008:**
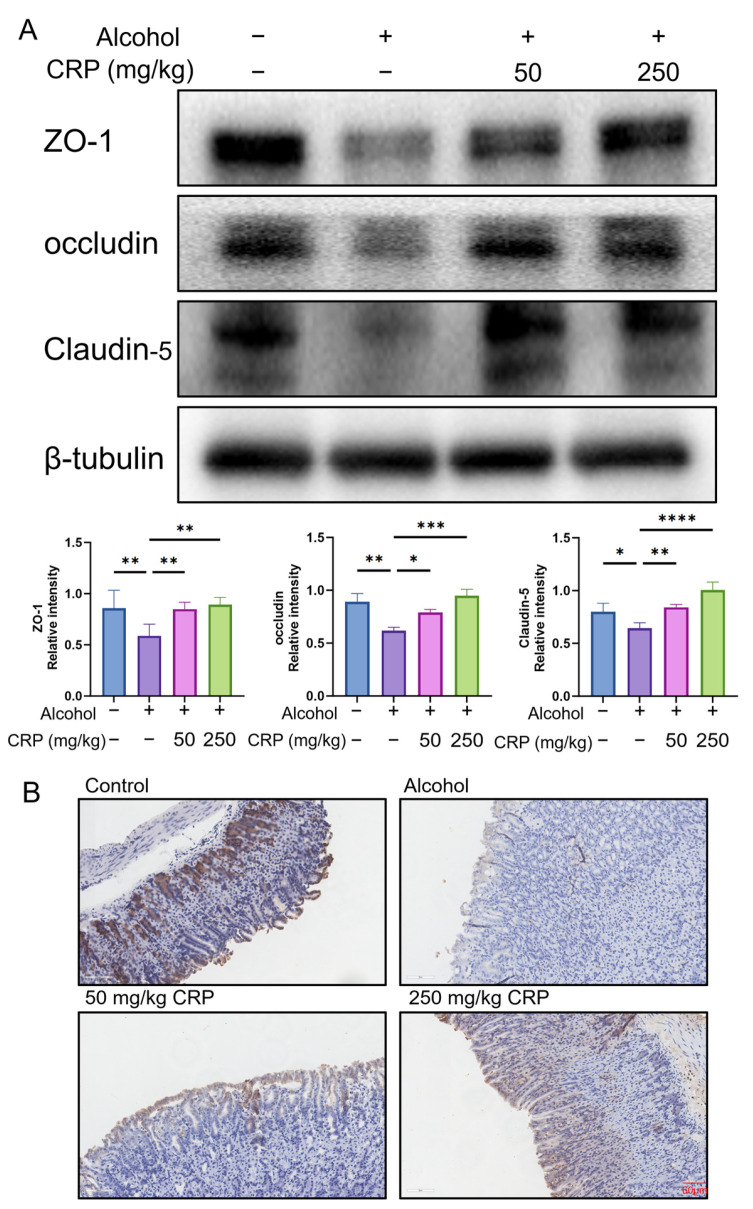
Effect of CRP pretreatment on TJs proteins. (**A**) Western blot investigation of the TJs protein expression prior to CRP pretreatment. β-tubulin was used as a loading control to ensure equal protein loading. Data are presented as means ± SD (n = 8). * *p* < 0.05, ** *p* < 0.01, *** *p* < 0.001, **** *p* < 0.0001 versus the alcohol-induced group. (**B**) Immunohistochemical analysis of ZO-1 expression in the gastric mucosa (scale bar: 60 μm).

**Figure 9 nutrients-17-02062-f009:**
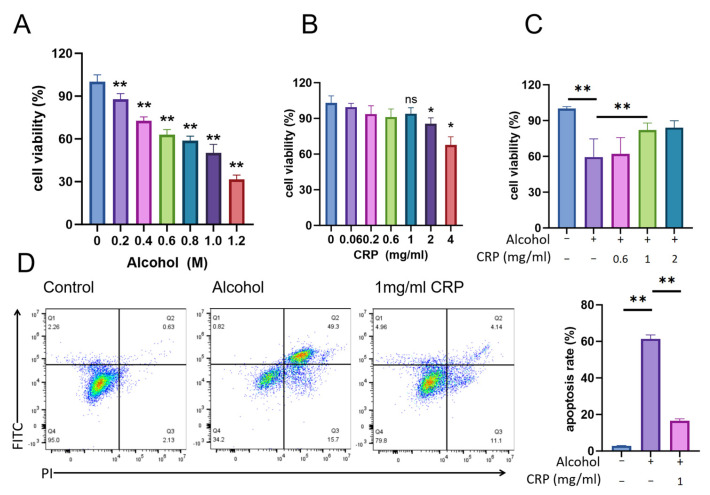
Effect of CRP pretreatment on alcohol-induced damage in GES-1 cells. (**A**) GES-1 cells were exposed to varying concentrations of alcohol, and cell viability was subsequently assessed. The results are presented as the mean ± SD (n = 9), and statistical analysis was conducted using one-way ANOVA. * *p* < 0.01, ** *p* < 0.01, compared with the control group. (**B**) GES-1 cells were exposed to varying concentrations of CRP, and cell viability was subsequently assessed. Statistical analysis was performed as in (**A**). (**C**) GES-1 cells were treated with different concentrations of CRP and 1 M alcohol for 4 h. The results are presented as the mean ± SD (n = 9), and statistical analysis was conducted using one-way ANOVA. ** *p* < 0.01, compared with the alcohol-induced group. (**D**) FITC-PI staining and Flow cytometric analysis of apoptosis. Statistical analysis was performed as in (**A**).

**Figure 10 nutrients-17-02062-f010:**
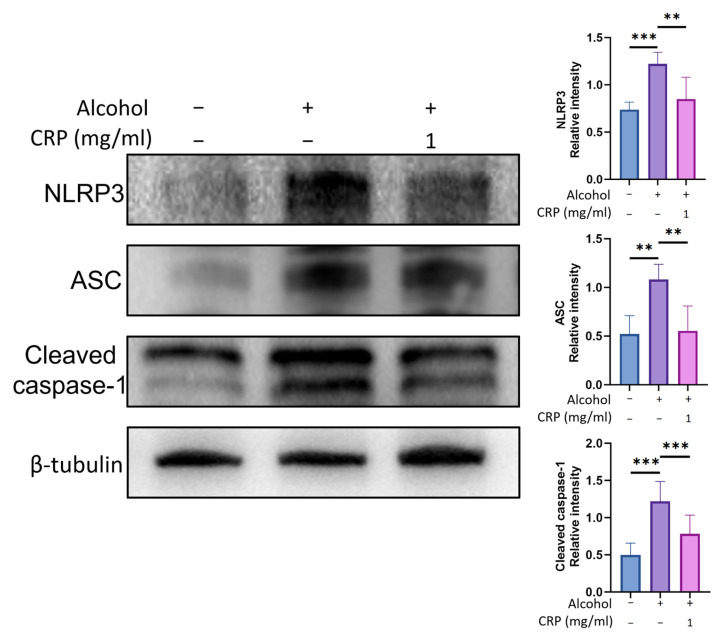
Effect of CRP on alcohol-induced NLRP3 inflammasome activation in GES-1 cells. Western blot results of NLRP3, ACS, and caspase-1 proteins. β-tubulin was used as a loading control to ensure equal protein loading. Data are presented as means ± SD (n = 8). ** *p* < 0.01, *** *p* < 0.001 versus the alcohol-induced group.

**Figure 11 nutrients-17-02062-f011:**
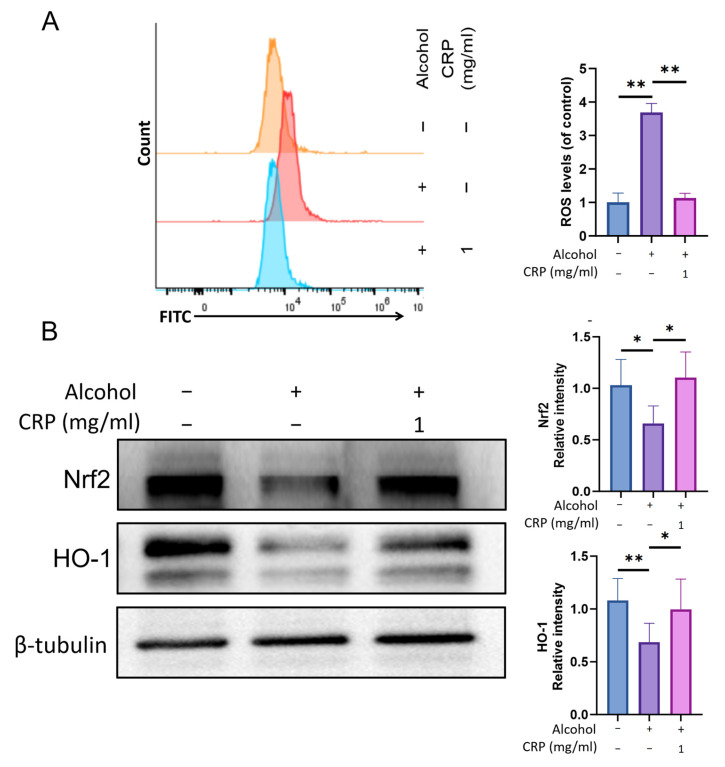
Effect of CRP on alcohol-induced oxidative stress in GES-1 cells. (**A**) ROS levels quantified by flow cytometry. (**B**) The protein expression levels of Nrf2 and HO-1 in gastric tissues were analyzed by Western Blot. β-tubulin was used as a loading control to ensure equal protein loading. Data are presented as means ± SD (n = 8). * *p* < 0.05, ** *p* < 0.01 versus the alcohol-induced group.

## Data Availability

All data associated with this study are available from the corresponding author upon reasonable request.

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
