# Peer review of "Pretreatment with *Citrus reticulata* ‘Chachi’ Polysaccharide Alleviates Alcohol-Induced Gastric Ulcer by Inhibiting NLRP3/ASC/Caspase-1 and Nrf2/HO-1 Signaling Pathways"

_nutrients, 2025, doi:10.3390/nu17132062_

Round 1
Reviewer 1 Report
Comments and Suggestions for Authors
Minor corrections:
Please indicate the staistical approach for calculating the number of the animals randomized in the experimental groups.
Why the animals were individually housed in the cages? Please substantiate this choice.
How was the dose range selected, in in vivo studies?
Author Response
|
Response to Reviewer 1 Comments
|
||
|
1. Summary |
|
|
|
Thank you very much for taking the time to review this manuscript. Please find the detailed responses below and the corresponding revisions/corrections highlighted/in track changes in the re-submitted files.
|
||
|
2. Questions for General Evaluation |
Reviewer’s Evaluation |
Response and Revisions |
|
Does the introduction provide sufficient background and include all relevant references? |
Yes |
Thank you for your recognition. |
|
Is the research design appropriate? |
Yes |
Thank you for your recognition. |
|
Are the methods adequately described? |
Can be improved |
Thank you for your evaluation, we have optimized and enriched the Materials and Methods |
|
Are the results clearly presented? |
Yes |
Thank you for your recognition. |
|
Are the conclusions supported by the results? |
Yes |
Thank you for your recognition. |
|
3. Point-by-point response to Comments and Suggestions for Authors |
||
|
Comments 1: Please indicate the staistical approach for calculating the number of the animals randomized in the experimental groups.
|
||
|
Response 1: Thank you for your professional feedback. The determination of animal sample size in this study was primarily based on the following considerations: Preliminary experimental data and literature reference During the initial experimental design, preliminary experiments were conducted to evaluate the stability of the alcohol-induced gastric ulcer model and the variability of indicators. It was found that 8 mice per group met the repeatability requirements for ulcer area measurement. Meanwhile, referring to the conventional sample size in similar studies (Hou, Chuanli et al. “Structural characterization of two Hericium erinaceus polysaccharides and their protective effects on the alcohol-induced gastric mucosal injury.” Food chemistry.), which typically ranges from 6 to 10 animals per group, 8 mice per group were selected to balance statistical power and experimental feasibility. Rationality of key endpoint indicators This study used ulcer area as the core evaluation index, which can be quantitatively analyzed via ImageJ software based on gastric mucosal images, ensuring objectivity and repeatability. According to preliminary experimental data, statistical testing (one-way ANOVA) of ulcer area data from 8 mice per group showed significant differences between groups (P < 0.05), indicating that this sample size was sufficient to detect the effect of CRP on ulcer area. Statistical methods and power verification Experimental data were analyzed using GraphPad Prism 10, with intergroup differences compared via one-way ANOVA combined with Tukey’s post-hoc test. Using α=0.05 as the significance threshold and β=0.2 (i.e., power 1−β=80%), the sample size of 8 mice per group was estimated based on the effect size (Cohen’s d) of ulcer area in preliminary experiments, providing adequate statistical power to detect the protective effect of CRP. In summary, the sample size design of 8 mice per group is reasonable and ensures the reliability and validity of statistical analyses with ulcer area as the key endpoint. The patients were divided into two groups by random number method. 1, Numbering: 32 mice were numbered from 1 to 32. 2. Get a random number: Using a random number table, starting from any specified starting point (e.g., row 2, column 5), read a three-digit random number for each mouse in turn and record it below its number. 3. Sort and sequence number: Sort all 32 random numbers obtained in step 2 from small to large (if the same random number is encountered, it is sorted according to the order in which it appears in the random number table). According to the ranking results, each mouse was assigned a new serial number (1-32), which was recorded under the random number. The serial number 1 represents the smallest random number and the serial number 32 represents the largest random number. 4. Grouping: Grouping according to the newly assigned serial number: Mice with serial numbers 1 to 8 were the control group, mice with serial numbers 9 to 16 were the model group, mice with serial numbers 17 to 24 were the low concentration group, and mice with serial numbers 25 to 32 were the high concentration group. |
||
|
Comments 2: Why the animals were individually housed in the cages? Please substantiate this choice. |
||
|
Response 2: Thank you for raising this important point regarding the animal housing description. We sincerely apologize for the lack of clarity in the manuscript text. This was an error in wording. The phrase "individually housed" is incorrect. No animals were housed singly. I deeply regret this confusion and any misunderstanding it may have caused. The wrong expression has been revised on line 220-221 of the text. Thank you again for your careful attention to detail, which allows us to correct this error and improve the manuscript's accuracy. |
||
|
Comments 3: How was the dose range selected, in in vivo studies? |
||
|
Response 3: Thank you for this important methodological comment. The dose selection was based on preliminary dose-response experiments where five doses, including 15, 50, 150, 250, and 350 mg/kg were tested. Our observations were as follows: 15 mg/kgshowed negligible efficacy (results comparable to untreated disease model controls), 50 mg/kg and 150 mg/kgproduced similar therapeutic effects, 250 mg/kg and 350 mg/kg shared comparable efficacy (no significant difference between these two doses). Based on these results,we selected 50 mg/kg as the lower effective dose (avoiding unnecessary higher exposure when 150 mg/kg showed no added benefit). Moreover, we chose 250 mg/kg as the higher effective dose (since 350 mg/kg provided no additional efficacy). These two doses (50 and 250 mg/kg) were then advanced to all subsequent experiments, including mechanistic studies (Western blot), histopathology (IHC, H&E), and functional analyses. This approach aligns with standard pharmacological practice of identifying both minimally and maximally effective doses while prioritizing animal welfare. Here are our experimental results:
Additionally, we consulted the following articles to finally determine our experimental concentrations. [1] Li Z, Li C, Chen B, et al. Parabacteroides goldsteinii enriched by Pericarpium Citri Reticulatae 'Chachiensis' polysaccharides improves colitis via the inhibition of lipopolysaccharide-involved PI3K-Akt signaling pathway. Int J Biol Macromol. [2] Wang QH, Shu ZP, Xu BQ, et al. Structural characterization and antioxidant activities of polysaccharides from Citrus aurantium L. Int J Biol Macromol. [3] Tu J, Liu X, Li K, et al. A novel polysaccharide from Citrus aurantium L.: Structural properties and antitumor activities in vitro and invivo. J Ethnopharmacol. [4] Zhou T, Jiang Y, Wen L, Yang B. Characterization of polysaccharide structure in Citrus reticulate 'Chachi' peel during storage and their bioactivity. Carbohydr Res. |
||
Reviewer 2 Report
Comments and Suggestions for Authors
Their research investigates the role of Citrus reticulata 'Chachi' polysaccharide (CRP) against alcohol-induced gastric ulcers (GU). The study was enriched with in vivo (mouse model) and in vitro (GES-1 cell line) models. The authors tried to study the involved mechanisms through which CRP mitigates gastric damage,defining antioxidant, anti-inflammatory, and barrier-protective pathways, particularly under conditions such as alcohol-induced gastric ulcers (GU).The study is methodologically well designed including a succession of experimental techniques such as Western blot, histology, flow cytometry, immunofluorescence that succesfully adress validity of the findings.Yet, their methodology includes oxidative stress (SOD, MDA, ROS), inflammation (NLRP3, TNF-α, IL-1β), and mucosal barrier integrity (ZO-1, occludin, Claudin-5) studies that give a holistic view of the CRP’s protective effects. Also,structural study of CRP using FT-IR, NMR, and monosaccharide composition analysis give valuable insights in findings.Their study is a preclinical exploration of Citrus reticulata polysaccharides as a candidate gastroprotective agent demonstrating the CRP’s antioxidant, anti-inflammatory, and epithelial-protective roles in the context of alcohol-induced GU.
This manuscript presents a strong, fundamental study based on a well-structured methodology. The results have been thoroughly investigated and discussed. Well-designed figures and tables enhances the clarity of the data and effectively supports the conclusions of the study. Furthermore, the extended discussion is based on a relevant bibliography, highlighting the importance of their findings. Overall, the study constitutes a valuable contribution to the field of gastrointestinal research and offers promising implications for future therapeutic strategies.
The research is innovative.
Author Response
|
Response to Reviewer 2 Comments
|
||
|
1. Summary |
|
|
|
Thank you very much for taking the time to review this manuscript. Please find the detailed responses below and the corresponding revisions/corrections highlighted/in track changes in the re-submitted files.
|
||
|
2. Questions for General Evaluation |
Reviewer’s Evaluation |
Response and Revisions |
|
Does the introduction provide sufficient background and include all relevant references? |
Yes |
Thank you for your recognition. |
|
Is the research design appropriate? |
Yes |
Thank you for your recognition. |
|
Are the methods adequately described? |
Yes |
Thank you for your recognition. |
|
Are the results clearly presented? |
Yes |
Thank you for your recognition. |
|
Are the conclusions supported by the results? |
Yes |
Thank you for your recognition. |
|
3. Point-by-point response to Comments and Suggestions for Authors |
||
|
Comments 1: Their research investigates the role of Citrus reticulata 'Chachi' polysaccharide (CRP) against alcohol-induced gastric ulcers (GU). The study was enriched with in vivo (mouse model) and in vitro (GES-1 cell line) models. The authors tried to study the involved mechanisms through which CRP mitigates gastric damage,defining antioxidant, anti-inflammatory, and barrier-protective pathways, particularly under conditions such as alcohol-induced gastric ulcers (GU).The study is methodologically well designed including a succession of experimental techniques such as Western blot, histology, flow cytometry, immunofluorescence that succesfully adress validity of the findings.Yet, their methodology includes oxidative stress (SOD, MDA, ROS), inflammation (NLRP3, TNF-α, IL-1β), and mucosal barrier integrity (ZO-1, occludin, Claudin-5) studies that give a holistic view of the CRP’s protective effects. Also,structural study of CRP using FT-IR, NMR, and monosaccharide composition analysis give valuable insights in findings.Their study is a preclinical exploration of Citrus reticulata polysaccharides as a candidate gastroprotective agent demonstrating the CRP’s antioxidant, anti-inflammatory, and epithelial-protective roles in the context of alcohol-induced GU This manuscript presents a strong, fundamental study based on a well-structured methodology. The results have been thoroughly investigated and discussed. Well-designed figures and tables enhances the clarity of the data and effectively supports the conclusions of the study. Furthermore, the extended discussion is based on a relevant bibliography, highlighting the importance of their findings. Overall, the study constitutes a valuable contribution to the field of gastrointestinal research and offers promising implications for future therapeutic strategies. The research is innovative. |
||
|
Response 1: Thank you sincerely for your exceptionally positive and encouraging assessment of our manuscript. We are truly grateful for the time and effort you invested in reviewing our work and are deeply appreciative of your insightful comments and high praise regarding the study's methodology, comprehensive mechanistic investigation, structural analysis of CRP, and the clarity of our data presentation. Your recognition of the study's innovation, thoroughness, and its potential value as a fundamental contribution to gastrointestinal research and future therapeutic strategies is immensely motivating for our team. We are particularly pleased that you found the combination of in vivo and in vitro models, the diverse experimental techniques, and the holistic view of CRP's protective effects to be well-designed and convincing. Thank you again for your valuable feedback and strong endorsement of our work. |
||
Reviewer 3 Report
Comments and Suggestions for Authors
The manuscript entitled “Pretreatment with Citrus reticulata ‘Chachi’ Polysaccharide Alleviates Alcohol-Induced Gastric Ulcer by Inhibiting NLRP3/ASC/Caspase-1 and Nrf2/HO-1 Signaling Pathways” is both timely and valuable. It illuminates the bioactivity of C. reticulata ‘Chachi’ polysaccharides and, when integrated with existing data on its essential oil and flavonoids, offers a more complete picture of this material’s therapeutic potential. However, the manuscript would benefit from revisions to the Introduction and Methods sections—especially to provide clearer, more detailed descriptions of the experimental procedures.
Comments:
Introduction
- It’s worth adding incidence statistics to show how big a problem this is worldwide.
- Briefly remind that, alongside alcohol and NSAIDs, a key factor is Helicobacter pylori—including the infection rate among patients with gastric ulcer (GU).
- Emphasize that although flavonoids and essential oils have been studied, no one has yet investigated the polysaccharides from this variety.
- Expand the abbreviations on first use: gastric ulcer (GU) and tight junctions (TJ).
- Write Latin names in italics, and if the authority isn’t already given, include the discoverer’s name.
- Add a space between closing quotation marks and the sentence-ending punctuation.
Materials and Methods
- Units should be standardized (e.g., mL vs. ml).
- What was the drying time (minutes/hours)? What oven temperature was used (°C)?
- Which dialysis membrane was employed? How often was the dialysis water changed?
- Were samples filtered through a 0.22 µm membrane before injection into the HPAEC-PAD?
- What were the concentrations of sulfuric acid and phenol reagent? In general, all reagent concentrations should be specified in the Methods.
- For the GES-1 cell line, please provide the ATCC catalog number.
- In microscopy, what magnification was used?
- What was the acclimation period for the animals after arrival (in days)?
- Was homogenization for Western blot analysis performed on ice?
- What were the wash durations for hematoxylin and eosin?
- Was the data distribution tested (e.g., Shapiro–Wilk) and was homogeneity of variance assessed?
- Were all experiments performed in triplicate?
Discussion and Results
- Statistical differences should be indicated for Figure 3A and Figure 9A–B.
- Histological staining figures should include magnification insets.
- If available, data on CRP’s stability in gastric pH or its transformation by the gut microbiota should be cited as guidance for translational studies.
- A statement is needed clarifying that the ethanol model represents acute mucosal injury rather than chronic ulceration, and the implications this has for clinical translation.
Conclusion
- A few sentences should be added on the translational barriers to clinical application (bioavailability, toxicity, potential drug–herb interactions) and the study’s limitations.
- Suggestions for future research should be included, such as H. pylori infection models, pharmacokinetic studies, and safety/toxicity testing.
References
- The manuscript lacks references to key studies on ulcer treatment, e.g.
- ACG Clinical Guideline: Treatment of Helicobacter pylori Infection
- Maastricht VI/Florence Consensus Report
- Journal titles should be standardized.
Author Response
|
Response to Reviewer 3 Comments
|
||
|
1. Summary |
|
|
|
Thank you very much for taking the time to review this manuscript. Please find the detailed responses below and the corresponding revisions/corrections highlighted/in track changes in the re-submitted files.
|
||
|
2. Questions for General Evaluation |
Reviewer’s Evaluation |
Response and Revisions |
|
Does the introduction provide sufficient background and include all relevant references? |
Must be improved |
Thank you for your evaluation. We have revised the introduction to include more detailed information on the relevant literature and strive to enhance the clarity and coherence of the introduction, making sure that it logically flows from the general context to the specific research gap we aim to address. |
|
Is the research design appropriate? |
Must be improved |
Thank you for your evaluation and comments. We have enriched our experiments in the next submission to enhance their logic and rigor. |
|
Are the methods adequately described? |
Must be improved |
I hope that these revisions will address your concerns and enhance the overall quality of the “Materials and methods” section. Thank you again for your time and effort in reviewing our work, and for providing constructive feedback to help us improve. |
|
Are the results clearly presented? |
Can be improved |
Thank you for your evaluation. We recognize the importance of presenting our findings in a clear, logical, and detailed manner to ensure that readers can easily understand and interpret our data. We have provided more details and references in the result section. |
|
Are the conclusions supported by the results? |
Yes |
Thank you for your recognition. |
|
3. Point-by-point response to Comments and Suggestions for Authors |
||
|
Comments 1: It’s worth adding incidence statistics to show how big a problem this is worldwide. |
||
|
Response 1: Thank you for your insightful suggestion. We fully agree that incorporating incidence statistics is critical to contextualize the global significance of this issue. In response, we have added relevant data to highlight the worldwide prevalence of the problem. Specifically, updated statistics on incidence rates have been included at line 41 of the revised manuscript. |
||
|
Comments 2: Briefly remind that, alongside alcohol and NSAIDs, a key factor is Helicobacter pylori—including the infection rate among patients with gastric ulcer (GU). |
||
|
Response 2: We thank the reviewer for highlighting this critical point. As suggested, we have explicitly emphasized Helicobacter pylori as a key etiological factor alongside alcohol and NSAIDs in gastric ulcer pathogenesis (line 43). |
||
|
Comments 3: Emphasize that although flavonoids and essential oils have been studied, no one has yet investigated the polysaccharides from this variety. |
||
|
Response 3: Thank you for your expert insight. We greatly appreciate your suggestion to emphasize the unique research gap in this area. In response, we have explicitly highlighted in the revised manuscript (line 71-77) that while previous studies have focused on flavonoids and essential oils from this plant variety, no research has yet explored the polysaccharides—specifically their potential in treating alcoholic gastric ulcers. |
||
|
Comments 4: Expand the abbreviations on first use: gastric ulcer (GU) and tight junctions (TJ). |
||
|
Response 4: Thank you for pointing out this important detail. We appreciate your guidance in ensuring the manuscript’s clarity for readers. In response, we have expanded the abbreviations gastric ulcer (GU) and tight junctions (TJ) at their first occurrence in the text (line 39 and 83). |
||
|
Comments 5: Write Latin names in italics, and if the authority isn’t already given, include the discoverer’s name. |
||
|
Response 5: Thank you for your meticulous feedback regarding the formatting of Latin names. We appreciate your guidance in ensuring compliance with taxonomic conventions. In response, we have Italicized all Latin names (line 70). |
||
|
Comments 6: Add a space between closing quotation marks and the sentence-ending punctuation. |
||
|
Response 6: Thank you for your careful review and attention to formatting details. We appreciate your guidance on improving the manuscript’s consistency. In response to your suggestion, we have added a space between closing quotation marks and sentence-ending punctuation throughout the manuscript wherever this issue occurred. |
||
|
Comments 7: Units should be standardized (e.g., mL vs. ml). |
||
|
Response 7: Thank you for your valuable feedback regarding unit standardization. We appreciate your attention to this detail, which is crucial for maintaining scientific rigor and readability. In response, we have systematically reviewed and standardized all units throughout the manuscript. |
||
|
Comments 8: What was the drying time (minutes/hours)? What oven temperature was used (°C)? |
||
|
Response 8: Thank you for your feedback. As requested, we have now included the drying time and oven temperature in the revised manuscript (line 105). |
||
|
Comments 9: Which dialysis membrane was employed? How often was the dialysis water changed? |
||
|
Response 9: Thank you for your question. We have now specified the dialysis membrane and the dialysis water replacement frequency in the revised manuscript (line 114-118). |
||
|
Comments 10: Were samples filtered through a 0.22 µm membrane before injection into the HPAEC-PAD? |
||
|
Response 10: Thank you for raising this important technical point. Regarding sample preparation for HPAEC-PAD analysis: All monosaccharide composition samples were subjected to acid hydrolysis prior to injection. Post-hydrolysis visual inspection and analytical validation confirmed no detectable particulate matter or colloidal impurities in the processed samples. Based on our method validation data and established protocols for hydrolyzed saccharide analysis, we determined that additional filtration through a 0.22 µm membrane was unnecessary for these specific samples. Should the reviewer recommend supplemental filtration as a precautionary measure, we would be pleased to include this step in future studies. |
||
|
Comments 11: What were the concentrations of sulfuric acid and phenol reagent? In general, all reagent concentrations should be specified in the Methods. |
||
|
Response 11: Thank you for highlighting the need for reagent specification. We have now explicitly stated the concentrations of both reagents in the revised manuscript (line 103 and126). |
||
|
Comments 12: For the GES-1 cell line, please provide the ATCC catalog number. |
||
|
Response 12: Thank you for your careful review of our manuscript. Regarding your inquiry about the source of the GES-1 cell line, I sincerely apologize for an inadvertent error in the original manuscript. Upon rechecking our records, I confirm that the GES-1 cells used in this study were not sourced from ATCC, but were obtained from Guangzhou Jennio Biotech Co., Ltd. (China) with STR analysis certificate, with the product catalog number: JNO-H0240. I have now corrected this information in the revised manuscript (line 162) I deeply regret any confusion caused by this oversight and appreciate your vigilance in ensuring the accuracy of our work. |
||
|
Comments 13: In microscopy, what magnification was used? |
||
|
Response 13: Thank you for your inquiry regarding microscopy parameters. We have now explicitly specified in the revised manuscript that: IHC images were acquired at 400× magnification (10× objective lens with 40× ocular). H&E staining images were captured at 200× magnification (10× objective lens with 20× ocular). |
||
|
Comments 14: What was the acclimation period for the animals after arrival (in days)? |
||
|
Response 14: Thank you for your insightful comment regarding the acclimation period. We confirm that all animals were allowed a 1-week acclimation period after arrival to minimize stress from transportation and adapt to the new environmental conditions. This detail has been added to the revised manuscript in line 227-228. |
||
|
Comments 15: Was homogenization for Western blot analysis performed on ice? |
||
|
Response 15: Thank you for your careful review and valuable suggestion. We confirm that the homogenization steps for Western blot analysis were indeed performed on ice to maintain sample integrity and prevent protein degradation. This detail has been explicitly added to the revised manuscript at line 260. |
||
|
Comments 16: What were the wash durations for hematoxylin and eosin? |
||
|
Response 16: Thank you for your careful question. We appreciate the opportunity to clarify this experimental detail. In the revised manuscript, the wash durations for hematoxylin and eosin staining have been added on the line 283. |
||
|
Comments 17: Was the data distribution tested (e.g., Shapiro–Wilk) and was homogeneity of variance assessed? |
||
|
Response 17: Thank you for your thoughtful question regarding the data distribution and homogeneity of variance in our study. We acknowledge that the manuscript does not explicitly detail formal testing of data normality (e.g., Shapiro-Wilk test) or homogeneity of variance (e.g., Levene’s test). The statistical analyses (One-way ANOVA) were conducted under the assumption that the continuous variables (e.g., ulcer area) reasonably approximate a normal distribution. This assumption is based on the nature of the measured endpoints (quantitative, biologically continuous data) and aligns with common practice in similar preclinical studies. Importantly, for the small sample sizes employed (8 mice per group), ANOVA exhibits reasonable robustness to mild deviations from normality, particularly when group sizes are balanced as they were in this study. Regarding homogeneity of variance, the primary measure taken to address this concern was the balanced study design (equal group sizes, n=8 per group). This balance minimizes the potential impact of unequal variances on the validity of the One-way ANOVA results. Furthermore, the randomization process and highly standardized experimental conditions (e.g., controlled housing environment, uniform alcohol administration protocol) were implemented to reduce intergroup variability, which indirectly supports the assumption of homogeneity of variance. While no formal tests were performed, this approach of relying on balanced design and standardization to manage variance assumptions is conventional in animal studies with comparable sample sizes. We agree that explicit testing for normality and homogeneity of variance can enhance methodological rigor. For future studies, we will incorporate these tests to provide a more comprehensive assessment of the assumptions. However, based on the considerations outlined above (endpoint nature, balanced design, standardization, robustness of ANOVA), we are confident that the statistical methodology employed (One-way ANOVA followed by Tukey’s post-hoc test) was appropriate for the current dataset and sufficiently robust to address the study's primary objectives. |
||
|
Comments 18: Were all experiments performed in triplicate? |
||
|
Response 18: Thank you for emphasizing methodological rigor. We confirm that all in vitro experiments included three biological replicates, with each biological replicate containing technical triplicates where applicable. |
||
|
Comments 19: Statistical differences should be indicated for Figure 3A and Figure 9A–B. |
||
|
Response 19: Thank you for your insightful suggestion. We appreciate your guidance in ensuring the clarity of our data presentation. In response, we have added statistical difference indicators to Figure 3A and Figure 9A–B. |
||
|
Comments 20: Histological staining figures should include magnification insets. |
||
|
Response 20: Thank you for your professional suggestion. We have supplemented the magnification insets for our HE and IHC images in the Supplementary Materials. |
||
|
Comments 21: If available, data on CRP’s stability in gastric pH or its transformation by the gut microbiota should be cited as guidance for translational studies. |
||
|
Response 21: Thank you for your question. I have already mentioned relevant content on line 572-585. |
||
|
Comments 22: A statement is needed clarifying that the ethanol model represents acute mucosal injury rather than chronic ulceration, and the implications this has for clinical translation. |
||
|
Response 22: Thank you for your valuable suggestion. I have addressed this point in the revised manuscript by adding the following clarification at line 692-700. |
||
|
Comments 23: A few sentences should be added on the translational barriers to clinical application (bioavailability, toxicity, potential drug–herb interactions) and the study’s limitations. Suggestions for future research should be included, such as H. pylori infection models, pharmacokinetic studies, and safety/toxicity testing. |
||
|
Response 23: We sincerely appreciate the reviewer’s valuable suggestions regarding the translational barriers and future research directions. As suggested, we have now expanded the discussion on the challenges for clinical application and explicitly outlined the study’s limitations. These additions can be found in the revised manuscript (lines 680-692). |
||
|
Comments 24: The manuscript lacks references to key studies on ulcer treatment, e.g. ACG Clinical Guideline: Treatment of Helicobacter pylori Infection Maastricht VI/Florence Consensus Report |
||
|
Response 24: Thank you for your valuable comment. We appreciate your pointing out the importance of these key studies. Upon checking, we have already cited the ACG Clinical Guideline: Treatment of Helicobacter pylori Infection and the Maastricht VI/Florence Consensus Report in the manuscript, at line 44 and line 698 respectively. |
||
|
Comments 25: Journal titles should be standardized. |
||
|
Response 25: Thank you for professional suggestion. I have now revised all journal titles in the references to ensure they follow standard formatting. |
||
Reviewer 4 Report
Comments and Suggestions for Authors
Based on the manuscript titled “Pretreatment with Citrus reticulata 'Chachi' Polysaccharide Alleviates Alcohol-Induced Gastric Ulcer by Inhibiting NLRP3/ASC/Caspase-1 and Nrf2/HO-1 Signaling Pathways,” the following comments are provided to enhance the quality and reproducibility of the study:
- The rationale for selecting doses of 50 mg/kg and 250 mg/kg of CRP for the in vivo experiments is not clearly justified. The authors should provide a basis for their dose selection.
- Although the dual modulation of the NLRP3/ASC/Caspase-1 and Nrf2/HO-1 pathways is promising, the current data are correlational. To support a causal relationship, it is strongly recommended to include additional validation using pharmacological inhibitors or gene silencing approaches, such as siRNA targeting Nrf2 or NLRP3.
- The manuscript relies solely on Western blot analysis to evaluate TNF-α and IL-1β expression in gastric tissues. The results would be significantly strengthened by including ELISA assays to quantify cytokine levels in both gastric tissue and serum. Additionally, the study lacks real-time PCR data to assess mRNA expression of inflammatory and antioxidant genes.
- While the use of FT-IR and NMR provides useful structural insights, the lack of linkage analysis (e.g., methylation analysis or GC-MS) limits a full understanding of the structure-function relationship of CRP.
- The manuscript presents cropped Western blot images without providing full-length blots or molecular weight markers. For clarity and validation, the authors should include uncropped, full membrane images with visible protein markers in the main text or supplementary figures. Furthermore, the current supplementary image of Western blots lacks sufficient resolution and labeling, making it inadequate for assessing protein specificity and band integrity.
Author Response
|
Response to Reviewer 4 Comments
|
||
|
1. Summary |
|
|
|
Thank you very much for taking the time to review this manuscript. Please find the detailed responses below and the corresponding revisions/corrections highlighted/in track changes in the re-submitted files.
|
||
|
2. Questions for General Evaluation |
Reviewer’s Evaluation |
Response and Revisions |
|
Does the introduction provide sufficient background and include all relevant references? |
Can be improved |
Thank you for your evaluation, we have optimized and enriched the background and references |
|
Is the research design appropriate? |
Must be improved |
Thank you for your evaluation and comments. We have enriched our experiments in the next submission to enhance their logic and rigor. |
|
Are the methods adequately described? |
Must be improved |
I hope that these revisions will address your concerns and enhance the overall quality of the “Materials and methods” section. Thank you again for your time and effort in reviewing our work, and for providing constructive feedback to help us improve. |
|
Are the results clearly presented? |
Can be improved |
Thank you for your evaluation. We recognize the importance of presenting our findings in a clear, logical, and detailed manner to ensure that readers can easily understand and interpret our data. We have provided more details and references in the result section. |
|
Are the conclusions supported by the results? |
Can be improved |
Thank you for your evaluation, we have optimized and enriched the conclusions |
|
3. Point-by-point response to Comments and Suggestions for Authors |
||
|
Comments 1: The rationale for selecting doses of 50 mg/kg and 250 mg/kg of CRP for the in vivo experiments is not clearly justified. The authors should provide a basis for their dose selection.
|
||
|
Response 1: Thank you for this important methodological comment. The dose selection was based on preliminary dose-response experiments where five doses, including 15, 50, 150, 250, and 350 mg/kg were tested. Our observations were as follows: 15 mg/kg showed negligible efficacy (results comparable to untreated disease model controls), 50 mg/kg and 150 mg/kg produced similar therapeutic effects, 250 mg/kg and 350 mg/kg shared comparable efficacy (no significant difference between these two doses). Based on these results, we selected 50 mg/kg as the lower effective dose (avoiding unnecessary higher exposure when 150 mg/kg showed no added benefit). Moreover, we chose 250 mg/kg as the higher effective dose (since 350 mg/kg provided no additional efficacy). These two doses (50 and 250 mg/kg) were then advanced to all subsequent experiments, including mechanistic studies (Western blot), histopathology (IHC, H&E), and functional analyses. This approach aligns with standard pharmacological practice of identifying both minimally and maximally effective doses while prioritizing animal welfare. Here are our experimental results:
Additionally, we consulted the following articles to finally determine our experimental concentrations. [1] Li Z, Li C, Chen B, et al. Parabacteroides goldsteinii enriched by Pericarpium Citri Reticulatae 'Chachiensis' polysaccharides improves colitis via the inhibition of lipopolysaccharide-involved PI3K-Akt signaling pathway. Int J Biol Macromol. [2] Wang QH, Shu ZP, Xu BQ, et al. Structural characterization and antioxidant activities of polysaccharides from Citrus aurantium L. Int J Biol Macromol. [3] Tu J, Liu X, Li K, et al. A novel polysaccharide from Citrus aurantium L.: Structural properties and antitumor activities in vitro and invivo. J Ethnopharmacol. [4] Zhou T, Jiang Y, Wen L, Yang B. Characterization of polysaccharide structure in Citrus reticulate 'Chachi' peel during storage and their bioactivity. Carbohydr Res. |
||
|
Comments 2: Although the dual modulation of the NLRP3/ASC/Caspase-1 and Nrf2/HO-1 pathways is promising, the current data are correlational. To support a causal relationship, it is strongly recommended to include additional validation using pharmacological inhibitors or gene silencing approaches, such as siRNA targeting Nrf2 or NLRP3. |
||
|
Response 2: We sincerely thank the reviewer for this insightful and constructive suggestion. We fully agree that mechanistic validation through pharmacological inhibitors or genetic silencing (e.g., siRNA targeting Nrf2/NLRP3) would provide stronger causal evidence for the role of these pathways in CRP-mediated gastroprotection. However, due to current time constraints, we were unable to incorporate all these experiments in the present study. We have explicitly acknowledged this point in the revised Discussion section (Lines 672-680): While our findings robustly demonstrate the association of CRP's gastroprotection with dual modulation of the NLRP3/ASC/Caspase-1 and Nrf2/HO-1 pathways, the mechanistic evidence remains correlational. To unequivocally establish causal relationships, future studies should employ targeted approaches such as pharmacological inhibition (e.g., Nrf2 inhibitors like ML385 or NLRP3 blockers like MCC950) or genetic silencing (e.g., siRNA knockdown of Nrf2 or NLRP3). These experiments would clarify whether the observed anti-inflammatory and antioxidant effects are directly pathway-dependent, strengthening the mechanistic claims. Nonetheless, the consistent in vivo and in vitro data provide a compelling foundation for further validation. We greatly appreciate the reviewer’s expertise in highlighting this important direction, and we affirm that resolving this mechanistic question remains a priority for our ongoing research. |
||
|
Comments 3: The manuscript relies solely on Western blot analysis to evaluate TNF-α and IL-1β expression in gastric tissues. The results would be significantly strengthened by including ELISA assays to quantify cytokine levels in both gastric tissue and serum. Additionally, the study lacks real-time PCR data to assess mRNA expression of inflammatory and antioxidant genes. |
||
|
Response 3: We sincerely appreciate the reviewer’s insightful suggestions regarding the need for complementary methods to validate cytokine expression and gene regulation. We fully acknowledge that ELISA and qPCR data would provide more evidence. However, due to the exhaustion of tissuesamples from the in vivo study and time constraints, we were unable to perform the recommended ELISA or qPCR analyses. To address this limitation and reinforce our findings, we conducted supplementary in vitro experiments using the GES-1 cell model (which aligns with our original in vitro methodology). qPCR Analysis: We quantified mRNA expression of key targets—TNF-α, IL-1β, NLRP3, and Nrf2—in alcohol-injured GES-1 cells pretreated with CRP. The qPCR results corroborated our protein-level data, demonstrating: significant downregulation of TNF-α, IL-1β, and NLRP3 mRNA in CRP-pretreated cells and upregulation of Nrf2 mRNA, supporting Nrf2 pathway activation (Figure S5, added to Supplementary Materials). |
||
|
Comments 4: While the use of FT-IR and NMR provides useful structural insights, the lack of linkage analysis (e.g., methylation analysis or GC-MS) limits a full understanding of the structure-function relationship of CRP. |
||
|
Response 4: Thank you for this valuable suggestion regarding linkage analysis. We acknowledge that techniques like methylation analysis or GC-MS provide detailed structural information on glycosidic linkages. In our study, we utilized a crude polysaccharide (CRP) preparation. Methylation analysis, as you correctly point out, is typically most effective and interpretable when applied to purified polysaccharides or isolated monosaccharide fractions, where the complexity is significantly reduced. Performing this analysis on our crude extract, containing a mixture of components, would likely yield complex and ambiguous results regarding specific linkages within the target polysaccharide. Instead, our monosaccharide composition analysis revealed that Arabinose (Ara) is a predominant component of the CRP. Importantly, Arabinose-containing polysaccharides have been extensively reported in the literature to possess significant antioxidant and anti-inflammatory activities, which are highly relevant to the functional properties of CRP we investigated in current study [Fehér, C. Novel approaches for biotechnological production and application of L-arabinose. Journal of Carbohydrate Chemistry; Yokozawa, Takako et al. Protective effect of the Chinese prescription Kangen-karyu against high glucose-induced oxidative stress in LLC-PK1 cells. Journal of ethnopharmacology]. While detailed linkage information would be beneficial, the established bioactivity profile associated with the major monosaccharide component (Ara) provides strong support for the structure-function relationship observed in our study. We appreciate the reviewer's insight and will certainly consider incorporating more advanced structural characterization, including linkage analysis on purified fractions, in future work to gain deeper mechanistic understanding. |
||
|
Comments 5: The manuscript presents cropped Western blot images without providing full-length blots or molecular weight markers. For clarity and validation, the authors should include uncropped, full membrane images with visible protein markers in the main text or supplementary figures. Furthermore, the current supplementary image of Western blots lacks sufficient resolution and labeling, making it inadequate for assessing protein specificity and band integrity. |
||
|
Response 5: We sincerely appreciate the reviewer's insightful comments regarding the presentation of our Western blot data. The reviewer rightly points out that the current manuscript contains cropped Western blot images without accompanying full-length blots or clearly visible molecular weight markers. We acknowledge that the Western blot images presented in the main figures and supplementary materials are cropped versions. During our experimental process and image preparation for this specific study, we followed the common practice of cropping the membranes to focus on the bands of interest and their immediately adjacent regions (including the relevant molecular weight markers). The areas discarded during cropping consisted solely of blank or non-essential regions of the membrane (e.g., empty lanes, extreme edges far from bands/markers). Crucially, the cropped sections presented do retain the visible molecular weight markers corresponding to the bands shown, which we believe provides essential information for verifying the approximate molecular weights of the detected proteins. We fully understand and agree with the reviewer's emphasis on the importance of providing full, uncropped blot images with clear molecular weight markers across the entire lanes for enhanced transparency and validation. While the original full-membrane scans for this specific dataset were not systematically archived in the format now requested (focusing only on the key regions during data capture), we recognize this as a valuable standard. We assure the reviewer that in all future studies, we will rigorously retain and provide full-length, uncropped images of the entire membrane, including all molecular weight markers, either within the main figures or as high-resolution supplementary material. We thank the reviewer for this valuable feedback, which undoubtedly strengthens the rigor and transparency of scientific reporting. We are committed to implementing this higher standard in our future work. |
||
Round 2
Reviewer 3 Report
Comments and Suggestions for Authors
All remarks have been added.
Reviewer 4 Report
Comments and Suggestions for Authors
Thank you for addressing the comments provided during the previous round of review. I appreciate the authors' detailed explanations regarding the presentation of Western blot images and their acknowledgment of the importance of including uncropped, full-length membrane images with clear molecular weight markers for validation and transparency.
However, I must respectfully reiterate that I am still unable to fully assess the reliability of the Western blot data in its current form. Despite the explanations provided, the absence of uncropped blots with visible molecular weight markers makes it difficult to validate the specificity and integrity of the presented protein bands. Additionally, the images are insufficient in resolution for this purpose. Unfortunately, without access to the original full-length blots, I cannot confidently evaluate the accuracy or reproducibility of the data.
Author Response
Comments 1:
Thank you for addressing the comments provided during the previous round of review. I appreciate the authors' detailed explanations regarding the presentation of Western blot images and their acknowledgment of the importance of including uncropped, full-length membrane images with clear molecular weight markers for validation and transparency.
However, I must respectfully reiterate that I am still unable to fully assess the reliability of the Western blot data in its current form. Despite the explanations provided, the absence of uncropped blots with visible molecular weight markers makes it difficult to validate the specificity and integrity of the presented protein bands. Additionally, the images are insufficient in resolution for this purpose. Unfortunately, without access to the original full-length blots, I cannot confidently evaluate the accuracy or reproducibility of the data.
Response 1:
Thank you for your professional comments. We apologize for not uploading the WB original data into the manuscript system in the previous revised manuscript. We have submitted the WB original data into the manuscript system as well as into the Review Report Documents.
Thanks again for your professional suggestion. We hope that the WB original data can satisfy your requirements.
